# A genetic screen identifies a protective type III interferon response to *Cryptosporidium* that requires TLR3 dependent recognition

**Alexis R. Gibson**[1], **Adam Sateriale**[1¤], **Jennifer E. Dumaine**[1], **Julie B. Engiles**[1,2], **Ryan D. Pardy**[1], **Jodi A. Gullicksrud**[1], **Keenan M. O'Dea**[1], **John G. Doench**[3], **Daniel P. Beiting**[1], **Christopher A. Hunter**[1], **Boris Striepen**[1]*

1 Department of Pathobiology, School of Veterinary Medicine, University of Pennsylvania, Philadelphia, Pennsylvania, United States of America, 2 Department of Pathobiology, New Bolton Center, School of Veterinary Medicine, University of Pennsylvania, Philadelphia, Pennsylvania, United States of America, 3 Genetic Perturbation Platform, Broad Institute of MIT and Harvard, Cambridge, Massachusetts, United States of America

¤ Current address: The Francis Crick Institute, London, United Kingdom
* striepen@upenn.edu

**Data Availability Statement:** Data are within the manuscript and supporting information files and are accessible through GEO accession number:

## Abstract

*Cryptosporidium* is a leading cause of severe diarrhea and diarrheal-related death in children worldwide. As an obligate intracellular parasite, *Cryptosporidium* relies on intestinal epithelial cells to provide a niche for its growth and survival, but little is known about the contributions that the infected cell makes to this relationship. Here we conducted a genome wide CRISPR/Cas9 knockout screen to discover host genes that influence *Cryptosporidium parvum* infection and/or host cell survival. Gene enrichment analysis indicated that the host interferon response, glycosaminoglycan (GAG) and glycosylphosphatidylinositol (GPI) anchor biosynthesis are important determinants of susceptibility to *C. parvum* infection and impact on the viability of host cells in the context of parasite infection. Several of these pathways are linked to parasite attachment and invasion and C-type lectins on the surface of the parasite. Evaluation of transcript and protein induction of innate interferons revealed a pronounced type III interferon response to *Cryptosporidium* in human cells as well as in mice. Treatment of mice with IFNλ reduced infection burden and protected immunocompromised mice from severe outcomes including death, with effects that required STAT1 signaling in the enterocyte. Initiation of this type III interferon response was dependent on sustained intracellular growth and mediated by the pattern recognition receptor TLR3. We conclude that host cell intrinsic recognition of *Cryptosporidium* results in IFNλ production critical to early protection against this infection.

## Author summary

*Cryptosporidium* infection is an important contributor to global childhood mortality. There are currently no vaccines available, and the only approved drug has limited efficacy in immunocompromised individuals and malnourished children who need it most. To

GSE185247. All code used to process and analyze the data is available through Code Ocean: https://doi.org/10.24433/CO.1074647.v1.

**Funding:** This work was supported in part by funding from the National Institutes of Health through grants to CAH and BS (R01AI148249), and fellowships and career awards to ARG (T32AI007532), AS (K99AI137442), JED (T32AI007532), and JAG (T32A1055400). The funders had no role in study design, data collection and analysis, decision to publish, or preparation of the manuscript.

**Competing interests:** The authors have declared that no competing interests exist.

discover which host proteins are essential for *Cryptosporidium* infection, we conducted a genome wide knockout screen in human host cells. Our results confirm the importance of glycosaminoglycans on the surface of epithelial cells for attachment and invasion of the parasite. We also found that host GPI anchor biosynthesis and interferon signaling pathways were enriched by our screen. Examining the role of interferon signaling further we found a type III interferon response, IFNλ, was generated in response to infection and shown to be initiated in the infected cell. Utilizing mouse models of infection, we found that the type III interferon response was important early during infection. We also determined that TLR3 was the pattern recognition receptor responsible for IFNλ production during *Cryptosporidium* infection. Our work shows that IFNλ acts directly on the enterocyte and its use in treating immunocompromised mice produced striking reductions in infection.

## Introduction

*Cryptosporidium* is a leading cause of diarrheal disease. In the United States, this apicomplexan parasite accounts for more than half of all waterborne disease outbreaks and infection can be life-threatening in individuals with compromised immune function [1,2]. Globally, the burden of this disease rests disproportionally on children under the age of two and the parasite is an important contributor to early childhood mortality [3]. Children can experience multiple episodes of infection, however, parasite and disease burden diminish over successive infection and non-sterile immunity protects children from severe illness as well as stunting [4].

It is well established that T cells are critical to protection from and the resolution of infection with *Cryptosporidium* [5]. The production of interferon gamma (IFNγ) is recognized to be one of the essential functions of T cells during *Cryptosporidium* infection [6], but T cells are not the only source of IFNγ [7–9]. Numerous other chemokines and cytokines produced by the enterocyte including IL-8, IL-18, TGFβ, and RANTES and type one and three interferons have been noted as well [10–14]. These can act directly on enterocytes and/or stimulate responses by proximal immune cells in the intestinal epithelium and adjacent tissues leading to the enhanced production of IFNγ, among other responses. IL-18 was shown to be produced by the enterocyte and to signal to ILC1s promoting IFNγ production [9]. New *in vitro* enteroid models of infection have also revealed the presence of a type I IFN response through RNA sequencing [15,16]. Additionally, type III interferon (IFNλ) production has been observed in response to *C. parvum* infection in neonatal piglets and neonatal mice [12]. Type III interferons, the most recently discovered members of the cytokine family, were shown to play unique roles at mucosal sites that could not be compensated for by type I interferons [17] making their role during *Cryptosporidium* infection of particular interest.

*Cryptosporidium* infection is typically restricted to the small intestine, but infection of the biliary tree and respiratory involvement has also been reported [18,19]. Within the intestine, the infection is limited to epithelial cells in which the parasite occupies an intracellular but extra-cytoplasmic niche at the brush border. A number of cytoskeletal and membranous structures separate the parasitophorous vacuole from the bulk of the infected enterocyte [20,21]. While reorganization of the actin cytoskeleton is one of the most prominent changes in host cell morphology, infection is also known to interfere with the composition and function of tight junctions, to induce tyrosine phosphorylation, and to activate PI3K signaling [22,23]. Recent studies have identified parasite proteins that are injected into host cell during and

after invasion [24,25] but we know very little about the specific components of the host cell that shape host-parasite interaction for *Cryptosporidium*.

Here, we used a CRISPR-Cas9 knockout screen to identify host genes that impact host cell survival during *Cryptosporidium* infection. The screen revealed the importance of several pathways, with IFN signaling, sulfated GAGs, and GPI anchor synthesis most prominent. We found that the interferon signaling pathway identified here was triggered by robust production of type III but not type I interferon in human host cells. This response required live infection and was initiated in infected cells. We investigated the molecular recognition mechanism that leads to this response and studied its impact on the infection. *In vivo* experiments showed IFNλ to limit parasite growth an effect that was independent of the presence of IFNγ. Thus, we elucidate a mechanism of cell intrinsic recognition and control of *Cryptosporidium*.

## Results

### A screen for host genes that impact *Cryptosporidium* infection and host cell survival

How *Cryptosporidium* interacts with its host cell is poorly understood. The parasite is thought to rely on pathogenesis factors exposed on its surface or secreted during and after invasion [26,27], however, host proteins are likely to play important roles in this interaction as well. To identify such host factors, we conducted an unbiased genetic screen. Since *Cryptosporidium* infection is limited to epithelial cells, we chose to screen in HCT-8 cells, a colon-derived human adenocarcinoma cell line widely used for experiments with this parasite [28]. In this *in vitro* culture system, parasites can only be propagated for 72 hours and then growth ceases. First, we measured the survival of HCT-8 cells over a range of infection conditions and found *C. parvum* to induce host cell death in a dose-dependent fashion over the 72 hours (Fig 1A). We chose to move forward with a 90% kill dose, and a multiplicity of infection (MOI) of 3, to impose strong selection for loss of function in genes required for parasite growth or cell death as part of the host response to infection. Next, we generated clonal HCT-8 cell lines that stably express Cas9 [29] and assessed activity in each clone using an EGFP reporter assay [30]. Briefly, Cas9 expressing cells were transfected with a lentiviral vector encoding EGFP as well as a single guide RNA (sgRNA) targeting the EGFP gene and analyzed by flow cytometry. Cells with Cas9 activity show reduced fluorescence when compared to the parental cell line and are shown normalized to a control cell line expressing no EGFP (Fig 1B). Clones C, I, and K showed high activity and served as three independent biological replicates in the subsequent screen. Using the Brunello lentiviral CRISPR library, we targeted the full complement of human protein coding genes with four sgRNAs each in addition to controls [31] for a total of 77,441 sgRNAs. $10^8$ cells of each clone were transduced with the library at an MOI of 0.4 to ensure each cell received only one sgRNA. Following seven days of puromycin selection, cells were expanded to 4 T175 flasks, achieving roughly 500-fold coverage and infected with *C. parvum* at a 90% kill dose. After 72 hours, the media was changed, and surviving cells were allowed to expand. Cells were exposed to *C. parvum* for a total of three rounds of infection and expansion to enrich for resistant host cells (Fig 1C). Genomic DNA was extracted from the input population as well as following each round of infection.

### Genetic screen reveals genes required for infection and host response

Deep sequencing of the integrated sgRNAs and comparison with the input population revealed the progressive enrichment of a subset of sgRNAs with each round of infection (Fig 1D). We note that infection of Cas9 expressing HCT-8 in the absence of the sgRNA library did not

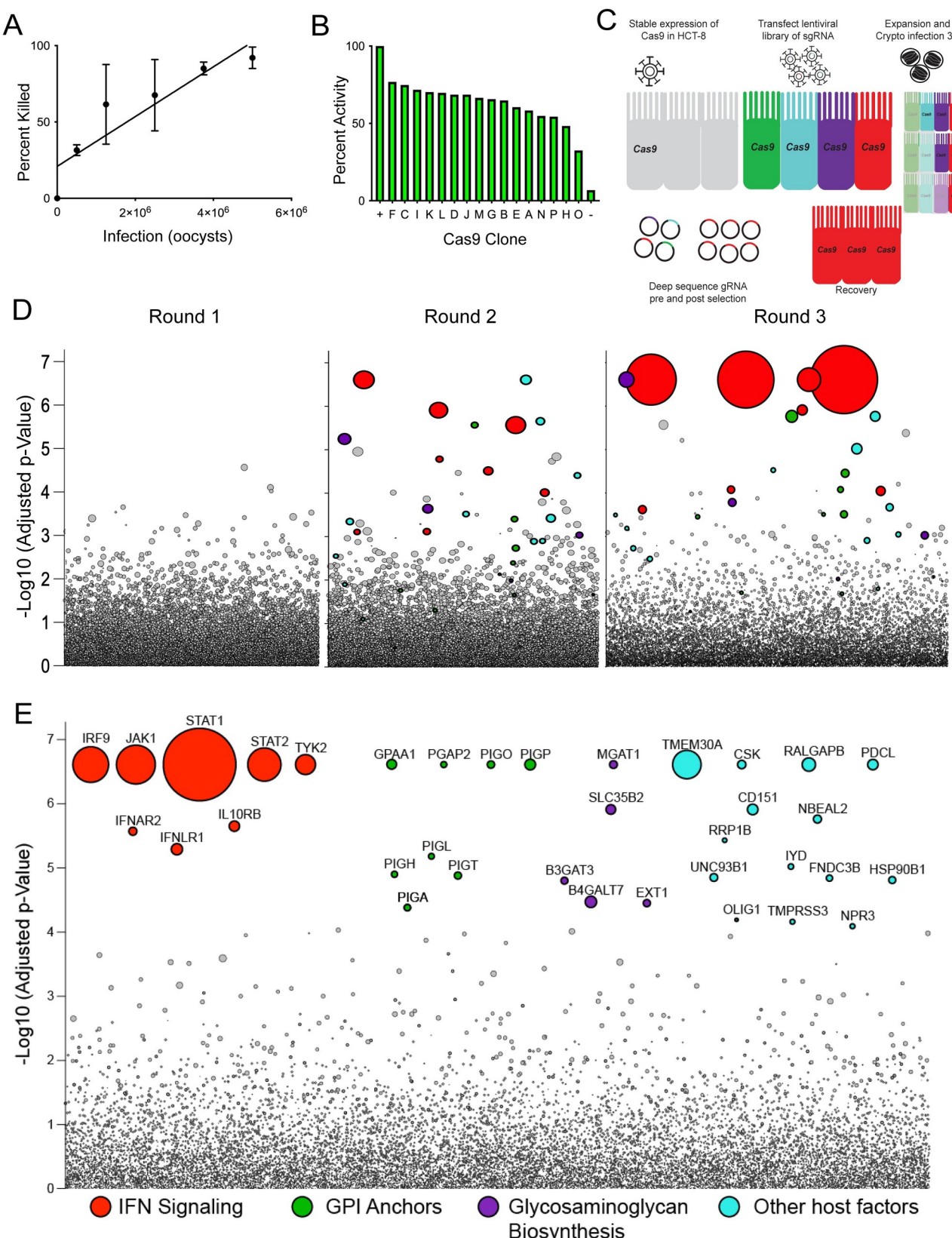

**Fig 1. A genome-wide screen reveals genes required for susceptibility to *Cryptosporidium* infection and host cell death.** To identify host genes required for *Cryptosporidium* infection we performed a genome-wide knockout screen. (A) HCT-8 cells infected with increasing numbers of *C. parvum* oocysts are killed in a dose-dependent manner. Host cell viability was assessed by Trypan Blue exclusion. $R^2$ = 0.7522. (B) Cas9 activity in different clones of Cas9 expressing HCT-8 cells assessed by flow cytometry normalized to a positive control set to 100 percent. (C) Schematic of CRISPR screen using *C. parvum* induced host cell death as selection. (D) Bubble plot of Cas9 expressing Clone K screen showing enrichment of specific genes with each round of selection by *C. parvum* infection. Each bubble represents a human gene and the size of bubbles corresponds to fold change. y-axis is the inverse of the adjusted *p* value. (E) Bubble plot of concatenation of Clones I and K comparing input to the final selection for 1000-fold coverage. The top 35 genes were colored and grouped based on function. Size of bubbles corresponds to fold change.

produce resistant host cells (S1 Fig). Clone I and Clone K were highly similar with consistent pathway enrichment across individual analyses (S3 Table). Clone C had poor sequencing depth and was therefore excluded from further analysis. Using model-based analysis of genome-wide CRISPR/Cas9 knockout [32], we identified 35 significantly enriched genes (FDR < 0.05, Fig 1E). Among these genes, gene set enrichment analysis (GSEA) revealed three distinct pathways each supported by multiple genes. IFNAR2, IFNLR1, IL10RB, IRF9, STAT1, STAT2, JAK1, and TYK2 cluster within the pathway of interferon (IFN) signaling. B3GAT3, B4GALT7, EXT1, MGAT1, and SLC35B2 are genes encoding enzymes in the biosynthesis of sulfated glycosaminoglycans (GAG). In addition, the screen selected for eight enzymes required for glycosylphosphatidylinositol (GPI) anchor biosynthesis (GPAA1, MGAT1, PGAP2, PIGA, PIH, PIGL, PIGO, PIGP, and PIGT). Beyond these pathways, a number of genes were significantly enriched that were not members of a particular pathway or represented the single representative of a pathway. Among those with known molecular function were accessory proteins to ATP flippase (TMEM30A), tyrosine protein kinases (CSK), GTPase activators (RALGAPB), G protein coupled receptor signaling regulators (PDCL), granule biogenesis proteins (NBEAL2), transcriptional activators of apoptosis (RRP1B), dehalogenases (IYD), tetraspanins (CD151), fibronectin domain proteins (FNDC3B), chaperones (UNC93B1 and HSP90B1), transcription factors (OLIG1), serine protease (TMPRSS3), and peptide hormone receptors (NPR3).

To validate the screening results, HCT-8 cells were transduced with siRNA targeting a subset of the top candidates for 24 hours prior to infection. Knockdown was assessed by qPCR and a decrease in transcripts was found to be typically 30% or greater. Cells were infected with a MOI of 0.8, which is lower than that used in the screen to account for the inability to transfect each cell. 48 hours after *C. parvum* infection, we assessed host cell viability using the MTT assay. We found that many candidates when knocked down, show increased resistance to cell death, no difference was noted in the absence of infection (S2 Fig).

### *Cryptosporidium parvum* infection induces an interferon response

Interferon signaling was the most highly enriched pathway identified by our screen. The critical role of IFNγ is well documented in humans [33] and mice [5] and there are also reports of *Cryptosporidium* associated induction of type I and III interferons [12,13,15]. To examine the epithelial cell response to *C. parvum*, we infected 6 well cultures of HCT-8 cells with 100,000 oocysts and performed RNA-seq. 1600 genes were differentially expressed (1.5-fold; adjusted *p* value < 0.05) by 48 hours post infection, compared to naïve cells (Fig 2A). The majority of differentially expressed genes were upregulated in the infected population compared to the uninfected control (689 genes downregulated). GSEA identified significant enrichment of the interferon signaling pathway in infected cultures compared to uninfected controls (Fig 2B). Other strongly enriched pathways are related to interferon signaling, such as REACTOME: Antiviral Mechanism by IFN Stimulated Genes.

To validate the observed interferon signature and to establish kinetics, we next conducted a qPCR time course experiment for three selected interferon stimulated genes (ISGs) over

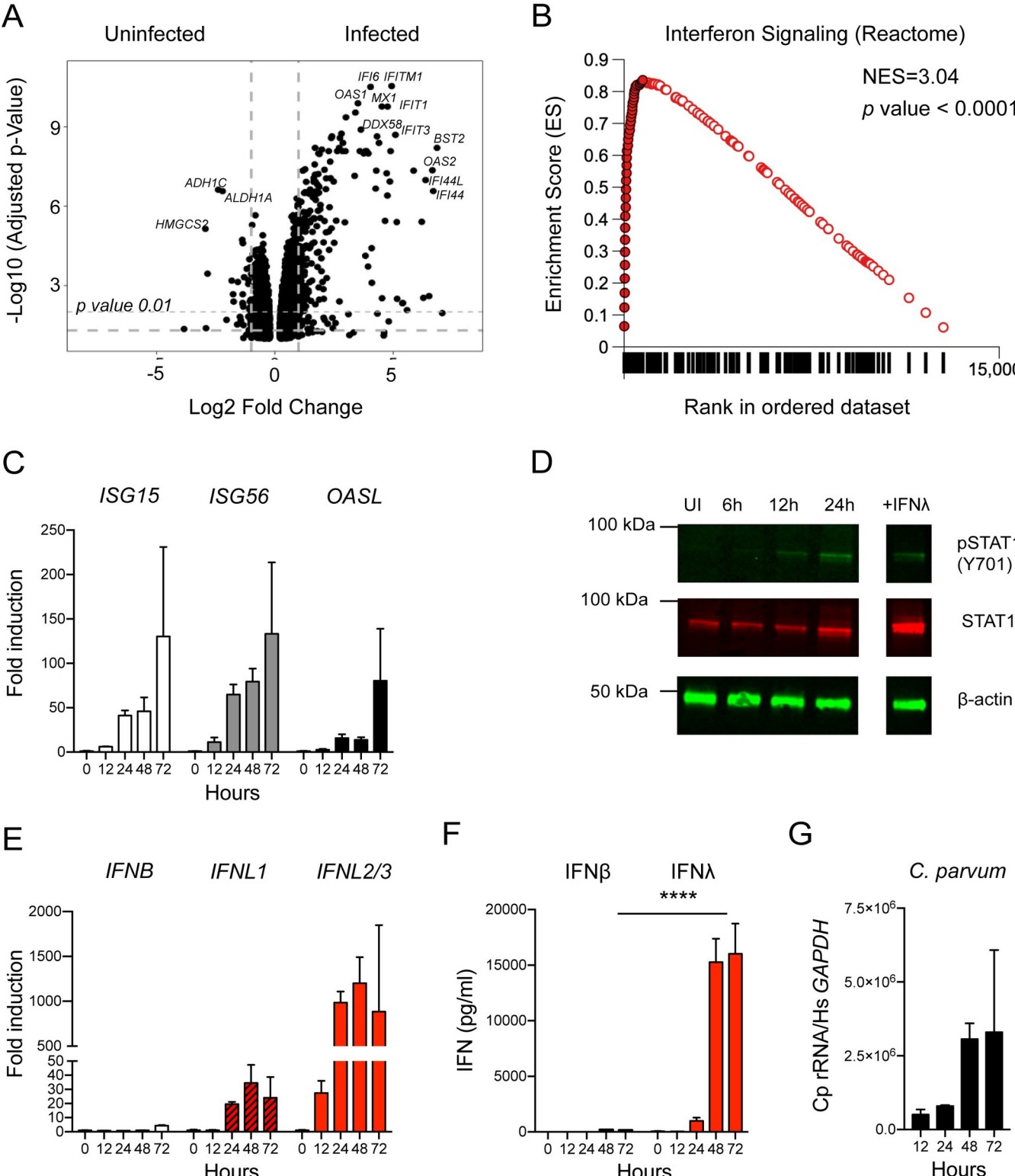

**Fig 2. *Cryptosporidium* infection induces a type III interferon response in human intestinal epithelial cells.** We examined the response to *C. parvum* infection in HCT-8 to determine which specific interferons were induced. (A) HCT-8 cultures were infected with *C. parvum* oocysts, and RNA was isolated 48 hours post infection from 3 biological replicates and matched uninfected controls. Volcano plot showing differentially expressed genes between uninfected and infected HCT-8 (n = 3, biological replicates per group). (B) GSEA plot of "REACTOME: Interferon Signaling" signature identified at 48 hours post infection. Closed circles represent genes that make up the core enrichment of the signature. Net enrichment score = 3.04, *p* value <0.0001. (C) 96 wells HCT-8 cultures

were infected with 25,000 *C. parvum* oocysts for 12–72 hours. Transcript abundance of three representative interferons stimulated genes (ISGs) measured by qPCR is shown over a time course of *C. parvum* infection (n = 3). (D) Immunoblot showing presence of phospho STAT1 and total STAT1 in uninfected cultures and following *C. parvum* infection. Treatment with IFNλ is used as a control for induction of phosphoSTAT1. One representative of 2 biological replicates is shown. (E) Samples as in (C). Induction of type I (IFNβ) and type III interferon (IFNλ) transcripts as assessed by qPCR. Note peak of *IFNL1*: 35-fold, *IFNL2/3*:1200-fold at 48 hours while peak *IFNB*: 4-fold at 72 hours. *n* = 3. (F) Protein levels of type I and type III interferons as assessed by ELISA. Samples as in (C, E) At 48 and 72 hours post infection, the difference between IFNβ and IFNλ was highly significant. Two-way ANOVA with Dunnett's multiple comparisons **** *p* <0.0001. *n* = 3. (G) Relative abundance of *C. parvum* ribosomal RNA transcripts normalized to host *GAPDH*. *n* = 3.

72hours of *C. parvum* infection to determine when the interferon response is initiated. We infected 96-well cultures with *C. parvum* and isolated RNA at 0, 12, 24, 48, and 72 hours post infection. ISG transcripts were increased in the first sample taken at 12 hours post infection and peaked at 72 hours (Fig 2C). Binding of interferons to their receptors initiates an intracellular signaling cascade that culminates in the phosphorylation of the transcription factor STAT1 leading to transcription of ISGs. We therefore assessed STAT1 phosphorylation by Western Blot using a modification specific antibody in whole cell lysates. Phosphorylation of STAT1 was not detectable in uninfected cells but was observed as early as 12 hours post infection (Fig 2D). We also observed an increase in total STAT1 protein at 24 hours post infection indicating that STAT1 itself was induced by the infection, in line with its classification as an ISG. We conclude that *C. parvum* infection induces a strong interferon response in HCT-8 cells.

## *Cryptosporidium* infection preferentially induces a type III interferon response

Next, we asked which interferon was responsible for the response observed. There are three major interferon types; type II interferon, IFNγ, is only produced by certain lymphocytes and thus absent from our cultures. In contrast, type I interferons, most prominently IFNα and IFNβ, and type III interferons, IFNλ 1–4, are known to be produced by epithelial lineages including the HCT-8 cells used here [34] (S2 Fig). Our GSEA analyses found enrichment signatures for type I and type III interferon in infected cultures (S2 Fig), but because both types act through the same intracellular signaling cascade, it is difficult to distinguish between them by the genes they induce [35]. To determine which types of interferons are expressed—simultaneously or individually—in response to *C. parvum*, we measured the transcript abundance of *IFNB*, *IFNL1*, and *IFNL2/3* by qPCR. *IFNB* transcripts did not increase at early time points and remained comparably low at 72 hours (4-fold, Fig 2E). In contrast, at 12 hours, the first time point sampled, type III interferon transcripts were already markedly elevated. Type III interferon transcripts peaked at 48 hours (*IFNL1*: 35-fold, *IFNL2/3*:1200-fold). We also performed enzyme-linked immunosorbent assays (ELISA) to directly measure protein levels for IFNβ and IFNλ. Only modest amounts of IFNβ were detectable, peaking at 48 hours post infection (185 pg/mL). IFNλ production was detected as early as 24 hours post infection and continued to increase until 72 hours, exceeding IFNβ levels by two orders of magnitude (16,029 pg/mL, *p* <0.0001, Two-way ANOVA, Fig 2F). The kinetics of the induction of IFNλ protein followed that of parasite replication, with a large increase between 24 and 48 hours, when the parasites were actively replicating, and a plateau between 48 and 72 hours when parasites terminally differentiate to gametes and growth ceases (Fig 2G and [36]). Taken together, these experiments demonstrate that type III, rather than type I interferons are preferentially induced by *C. parvum* infection in HCT-8 cells.

## Live *C. parvum* infection is required to induce type III interferon production

A variety of pathogen associated molecular patterns (PAMPs) have been shown to induce a type III interferon response including many bacterial proteins, glycans and lipids [37]. Oocysts

used in our experiments were isolated from the feces of cows or mice; therefore, we considered that inadvertent inoculation of cultures with bacterial PAMPS rather than *C. parvum* infection may drive IFNλ production. To test this, we heated oocysts to 95°C for 10 min prior to adding them to cells. This kills the parasite but does not inactivate LPS [38]. Heat killed parasites failed to induce IFNλ at any timepoint assessed, and at 72 hours post infection, the difference in IFNλ production compared to controls was highly significant (Fig 3A, $p < 0.0001$, Two-way ANOVA). Consistent with the lack of IFNλ production, we did not observe phosphorylation of STAT1 in cultures inoculated with heat killed parasites (Fig 3B). To further assess the importance of parasite replication for interferon induction, we used nitazoxanide, the only currently FDA approved drug for the treatment of *Cryptosporidium* infection. Treatment of cultures led to a 35-fold decrease in parasite infection as assessed by qPCR (Fig 3C). In the nitazoxanide treated infected cultures, IFNλ induction was no longer observed (Fig 3D, $p < 0.05$, One-way ANOVA). In contrast, the induction of IFNλ using an agonist of interferon signaling, Poly(I:C), was intact under nitazoxanide treatment, demonstrating that the observed response is specific to parasite infection. We therefore conclude that live parasites and active parasite replication are required to induce the type III interferon response.

## IFN-lambda is initially produced by infected cells and signals in an autocrine manner

The parasite completes its replicative cycle within 12 hours and parasite egress is accompanied by host cell lysis and the release of intracellular contents, including both host and parasite molecules (Fig 3E). Both intracellular parasite growth, and/or host cell lysis could trigger the interferon response. Furthermore, signaling, once initiated, results in the secretion of interferons, which may act on both producing and surrounding cells in an auto- as well as paracrine fashion. This amplifies the signal through a feedforward loop rapidly leading to cytokine from essentially all cells, making it difficult to determine how the cascade originates.

To determine the cells that initiate the type III interferon response, we infected HCT-8 with a *C. parvum* strain marked by expression of tandem Neon green fluorescent protein (Fig 3F). At 10 hours post infection, and prior to first parasite egress, we sorted cells for green fluorescence and isolated Neon positive infected cells as well as Neon negative bystander cells from the same culture (Fig 3G). Three biologically independent samples were subjected to RNA sequencing for each population. Infection resulted in significant differences in gene expression with 380 upregulated and 466 downregulated genes (1.5-fold; adjusted *p* value < 0.05, Fig 3H). We noted the induction of IFNL1 and 126 additional ISGs as identified by Interferome DB [39]. Many of these genes represent a subset of the interferon signature we observed in our 48-hour RNA-seq but the amplitude of expression was lower, likely a reflection of the early timepoint and the lack of paracrine amplification. Importantly, at this timepoint induction of the interferon pathway is exclusive to infected Neon positive cells. Notably, when compared to an uninfected control, bystander Neon negative cells do not show enrichment of the interferon signature (S5 Fig). We conclude that the type III interferon response is initiated during intracellular replication of *C. parvum* in a cell intrinsic fashion.

## The type III interferon response is required for early in vivo host defense

To understand the consequences of the type III interferon response on infection, we turned to an *in vivo* model of infection that uses a *C. parvum* strain adapted to mice by continued serial passage [40]. First, we asked whether and when type III interferons are produced in response to *C. parvum in vivo*. At day 2 post infection of C57BL/6 mice, we found an average 4-fold increase of *Ifnl2/3* transcripts in the small intestine, and at day 4, the induction was

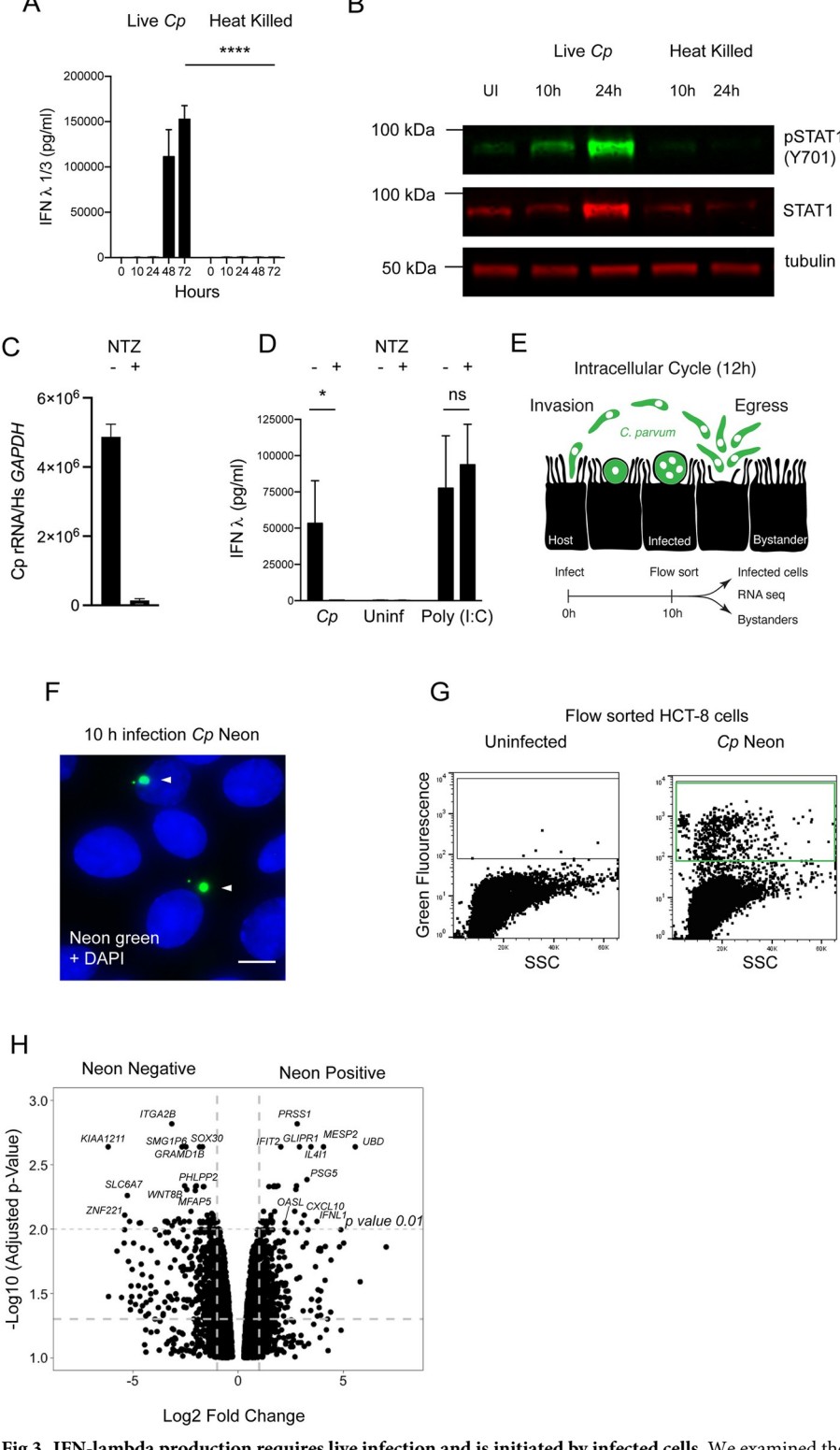

**Fig 3. IFN-lambda production requires live infection and is initiated by infected cells.** We examined the requirements and kinetics of initiation of the type III interferon response to *C. parvum*. (A) 24 well HCT-8 cultures were infected with 200,000 *C. parvum* live or heat killed oocysts and protein levels of IFNλ were assessed by ELISA. At 48 and 72 hours post infection the difference between live and heat killed was highly significant. Two-way ANOVA with Šídák's multiple comparisons test **** $p < 0.0001$. $n = 3$. (B) Immunoblot comparing induction of STAT1

phosphorylation by infection with live versus heat killed parasites. Phospho STAT1 is only detected in live infection. One representative example of two biological replicates is shown. (C) Following treatment with nitazoxanide (NTZ), infection is reduced 35-fold as assessed by relative abundance of *C. parvum* ribosomal RNA transcripts normalized to host *GAPDH*. $n$ = 3. (D) Protein levels of IFNλ in HCT-8 infected with *C. parvum* compared to cultures stimulated with 10μg/mL lipofected Poly(I:C), both in the presence or absence of nitazoxanide. A decrease in IFNλ protein is observed only when NTZ is used in infection. One-way ANOVA with Šídák's multiple comparisons test * $p$ <0.05. $n$ = 3. (E) Schematic of the 12-hour intracellular cycle of *C. parvum* and outline of a sequencing experiment to examine transcriptional differences between bystanders and infected cells from the same culture. (F) Immunofluorescence of HCT-8 infected with *Cp* Neon (green) at 10 hours post infection. Hoechst in blue. Scale bar 10μm. (G) Flow cytometry dot plot of infected cells showing green fluorescence and side scatter. Three biological replicates were sorted for Neon positive to Neon negative comparison. (H) Volcano plot showing differentially expressed genes between Neon negative (bystander) and Neon positive (infected) HCT-8 at 10 hours post infection.

approximately 2-fold. (Fig 4A). We also assessed IFNλ secretion during *C. parvum* infection using an ELISA from punch biopsies of the ileum and found similar kinetics. IFNλ secretion was increased at 2 days post infection and waned below detectable levels after 4 days (Fig 4B).

Type I and III interferons initiate a similar intracellular signaling cascade but utilize different receptors, a heterodimer of IFNAR1 and IFNAR2 for type I and a heterodimer of IFNLR1 and IL10RB for type III. We infected C57BL/6 wild type mice, mice lacking the type I interferon receptor, Ifnar1$^{-/-}$, and mice lacking the type III interferon receptor, Ifnlr1$^{-/-}$, with 50,000 *C. parvum* oocysts. Infection was monitored by measuring parasite produced Nanoluciferase from feces [41]. Surprisingly, loss of the type I interferon receptor consistently resulted in a 3-fold reduction in shedding when compared to wild type mice (area under the curve (AUC), Fig 4C, One-way ANOVA **** $p$ <0.0001). In contrast, loss of type III interferon signaling resulted in an overall 2.7-fold increase in parasite shedding compared to wild type mice (AUC, Fig 4C, One-way ANOVA **** $p$ <0.0001). We note that histopathology revealed no baseline differences between WT and Ifnlr1$^{-/-}$ mice (S7 Fig). To further validate this finding independent of mouse mutants, we used antibody-based depletion. C57BL/6 mice were intraperitoneally injected daily with 20μg of an anti-Ifnλ2/3 antibody and infected with 50,000 *C. parvum* oocysts. Again, we observed an increase in parasite shedding of about 2-fold (AUC, Fig 4D, Standard t-test **** $p$ <0.0001). We conclude that type III, but not type I interferons contribute to the early control of *Cryptosporidium in vivo*.

## Exogenous IFN-lambda treatment protects mice from severe *Cryptosporidium* infection

Since mice lacking the type III interferon receptor exhibited an increase in early susceptibility, we tested the impact of exogenous administration of IFNλ on *Cryptosporidium* infection. Here we use mice lacking IFNγ as they exhibit 100-fold higher infections than that observed in wild type [42], providing a greater dynamic range to study the utility of exogenous IFNλ treatment. Ifng$^{-/-}$ mice were injected intraperitoneally with daily doses of Ifnλ2 ranging from 0–5μg for the first three days of infection. As little as 0.1μg per day, the smallest amount tested, produced a marked reduction in shedding (4.3-fold decrease AUC, Fig 4E, One-way ANOVA ** $p$ <0.01), and increasing the dose beyond 1μg did not yield further enhancement. To assess whether this effect could be maintained long term, Ifng$^{-/-}$ mice were infected with *C. parvum* and injected intraperitoneally with a daily dose of 1μg Ifnλ2 for the duration of the infection. This treatment resulted in 7.7-fold reduction of shedding when compared to mock injected control infections (AUC, Fig 4F, Standard t-test * $p$ <0.05). On day 14 post infection, 100% mortality was observed in mock injected mice while no mortality was observed in treated mice. Mice lacking the type I IFN receptor were not more susceptible to infection but we wondered if type I IFN provided exogenously would impact *Cryptosporidium* infection. Treatment

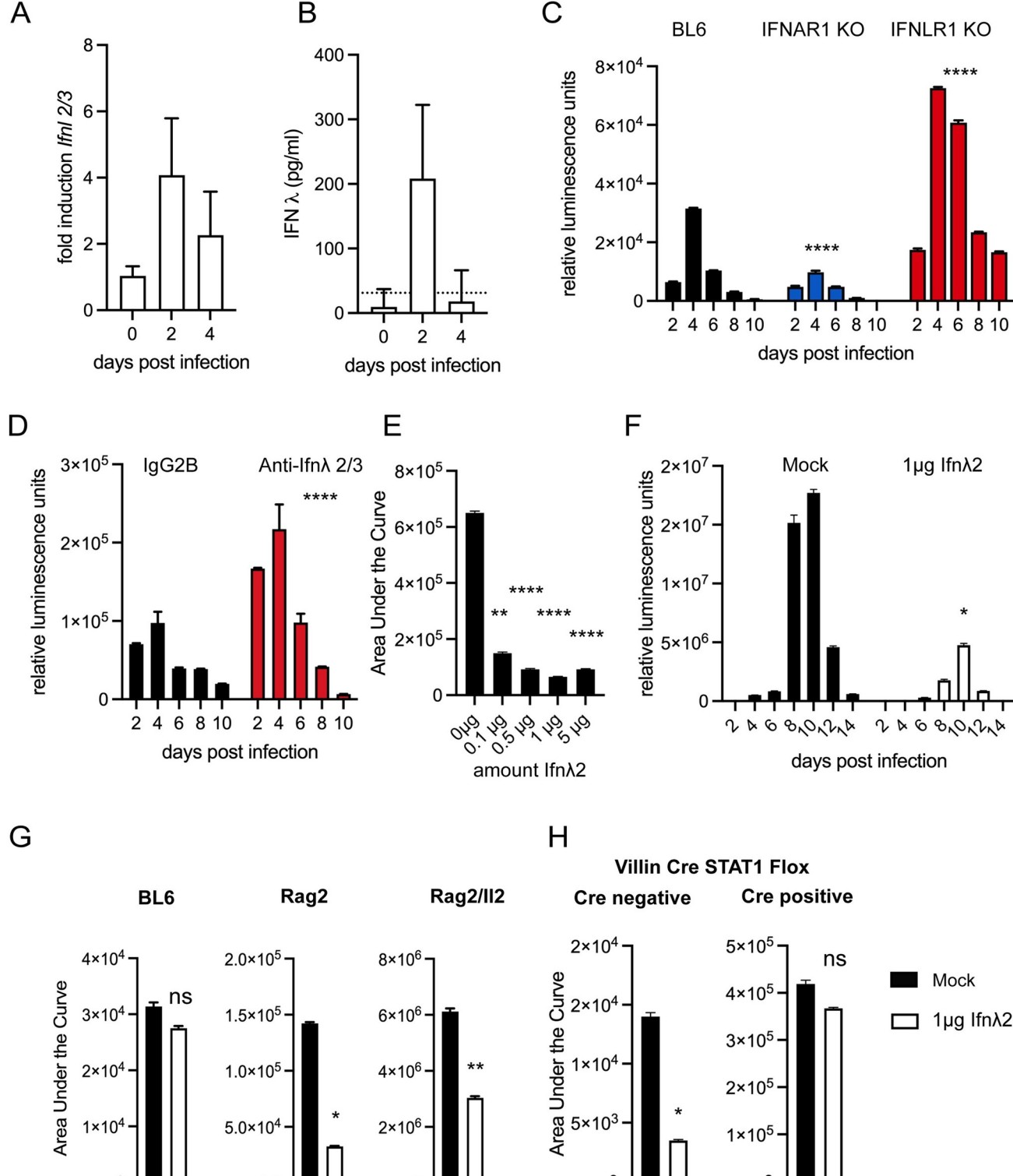

**Fig 4. The type III Interferon response is host protective and epithelial cell intrinsic.** We used a mouse model of infection to examine the role of type III interferon in *Cryptosporidium* infection *in vivo*. All mice were infected with mouse adapted *C. parvum*. (A) C57/BL6 (BL6) mice were infected with 50,000 *C. parvum* oocysts and relative abundance of IFNλ transcript is shown for ileal biopsies of infected mice, 8–10 mice per day, *n* = 3 independent biological replicates. (B) BL6 mice were infected with 50,000 *C. parvum* oocysts and secreted IFNλ protein from ileal biopsies was assessed by ELISA, 8–10 mice per day, *n* = 3 independent biological replicates. Dotted line represents the limit of detection. (C) Fecal luminescence measured every two days

in BL6 wild type mice, mice lacking the type I IFN receptor Ifnar1$^{-/-}$, and mice lacking the type III interferon receptor Ifnlr1$^{-/-}$ following infection with 50,000 *C. parvum*. A reduction of 3-fold was observed in Ifnar1 and an increase of 2.7-fold was observed with Ifnlr1. 4 mice per group. Data shown is representative of 3 biological replicates. (Ifnar1: -2.8, -2.9-fold, Ifnlr1: 2.3, 2.7-fold) 3–4 mice per group. n = 3 independent biological replicates. One-way ANOVA with Dunnett's multiple comparisons of area under the curve across all biological replicates **** $p < 0.0001$. (D) C57/BL6 mice were treated with anti-Ifnλ2/3 antibody or an isotype control daily via intraperitoneal (i.p.) injection and infected. 4 mice per group. Fecal luminescence was measured every two days. Representative of two biological replicates. An increase of 2-fold was observed in each replicate. *n* = 2. Standard *t*-test of area under the curve across two biological replicates **** $p < 0.0001$. (E) Ifng$^{-/-}$ mice were injected i.p. with indicated doses of Ifnλ2 daily for days 0–3 of infection. Mice were infected with 20,000 *C. parvum* oocysts. The total area under the curve of fecal luminescence for the 3-day infection is shown. 2 mice per dose. Representative of 2 biological replicates (0.1μg: 4.3 and 5.9-fold; 0.5μg: 7 and 1.3-fold; 1μg: 3.8 and 8.8-fold; 5μg: 7 and 8.6-fold). *n* = 2. One-way ANOVA with Dunnett's multiple comparisons of area under the curve across all biological replicates ** $p < 0.01$ **** $p < 0.0001$. (F) Ifng$^{-/-}$ mice were injected i.p. with 1μg of Ifnλ2 beginning at day 0 and each day for the duration of the infection. Mice were infected with 20,000 *C. parvum* oocysts. Fecal luminescence measured every two days. A 7.7-fold decrease in shedding occurred upon treatment representative of 2 biological replicates (5-fold decrease) 4–5 mice per group. *n* = 2. Standard *t*-test of area under the curve across two biological replicates * $p < 0.05$. (G) Wild type mice (B6), mice lacking T cells (Rag2$^{-/-}$), and mice lacking NK cells, ILCs, and T cells (Rag2/Il2rg$^{-/-}$) were treated with 1μg of Ifnλ2 daily for the days 0–3 of infection. The total area under the curve of fecal luminescence for the 3-day infection is shown. (BL6: 2.4, 1.14, 1.22-fold; Rag2$^{-/-}$ 4.4, 1.8, 16.3-fold; Rag2/Il2rg$^{-/-}$ 2, 3.8, 6.5-fold). Representative of 3 biological replicates. *n* = 3. Standard *t*-test of area under the curve across all biological replicates * $p < 0.05$, ** $p < 0.01$. (H) Villin Cre STAT1 flox mice or littermate Cre negative controls were treated with 1μg of Ifnλ2 daily for the days 0–3 of infection. The total area under the curve of fecal luminescence for the 3-day infection is shown. 2–3 mice per group. Representative of 2 biological replicates (Cre negative: 4 and 2.2-fold; Cre positive: 0.85 and 1.14-fold) *n* = 2 Standard *t*-test of area under the curve across two biological replicates * $p < 0.05$.

of infected Ifng$^{-/-}$ mice with IFNβ reduced infection by 2–3 fold (AUC, S8 Fig, Standard t-test * $p < 0.05$) and thus was less effective than IFNλ mirroring previous studies using rotavirus [43].

We note that administration of IFNλ was protective in Ifng$^{-/-}$ mice, suggesting that this protection does not require IFNγ. However, IFNλ has been shown to promote IFNγ production [44]. To examine this potential interaction further we tested the effect of IFNλ in mice lacking cells known to produce IFNγ in response to *C. parvum*: T cells, NK cells and ILCs [7,40,45]. BL6, Rag2$^{-/-}$, and Rag2$^{-/-}$Il2rg$^{-/-}$ mice were infected and treated with 1μg of Ifnλ2. This resulted in comparable reduction of parasite shedding (BL6: 1.14-fold, Rag2$^{-/-}$: 4.4-fold, Rag2$^{-/-}$Il2rg$^{-/-}$: 2-fold, Fig 4G), again suggesting that the benefit of IFNλ treatment does not require immune cells, but largely rests on an enterocyte intrinsic response. Finally, we conducted experiments with mice in which the STAT1 gene was specifically ablated from enterocytes using Cre recombinase under the control of the *Villin1* promoter. Villin Cre STAT1 floxed mice exhibit infections on the order of Ifng$^{-/-}$ mice due to the inability of the enterocyte to respond to IFNγ [9]. Removing STAT1 from the enterocyte lineage alone abolished the benefit of IFNλ treatment (1.1-fold AUC, Fig 4H, Standard t-test ns $p > 0.05$). Taken together, these data suggest that IFNλ protects mice and does so by acting directly on the intestinal epithelium.

## TLR3 detects *Cryptosporidium* infection leading to IFN-lambda production

Enterocytes have been shown to use a range of pattern recognition receptors to detect infection with different pathogens, many of which can lead to a type III interferon response [46]. HCT-8 cells, as many other cancer-derived lines, no longer express the full complement of innate immune recognition and cell death pathways [47], but the IFNλ response to *Cryptosporidium* remains intact. We took advantage of this to narrow the list of potential receptors. HCT-8 cells were treated with known agonists of different pattern recognition receptors and IFNλ production was measured after 24 hours. Specifically, we tested Poly(I:C) with or without lipofection (TLR3 and RLRs), mTriDAP (NOD1 and NOD2), 5' triphosphate dsRNA (RIGI), HSV60 DNA (CDS), ssPolyU RNA (TLR7), or CpG motif containing DNA (TLR9) [48]. As shown in Fig 5A, only Poly(I:C) and ssPolyU RNA produced an IFNλ response. This suggests TLR3, TLR7, or the RLRs MDA5 and RIGI as potential receptors. We next tested each of these candidates *in vivo* using suitable mouse mutants. Mice lacking MAVS, the adapter protein to RLRs, and TLR7 showed no difference in infection compared to wild type controls (Fig 5B and 5C).

However, mice lacking TLR3 were more susceptible, resulting in an 8-fold increase in parasite shedding and an overall pattern of infection that was reminiscent of Ifnlr1$^{-/-}$ mice (AUC, Fig 5D, Standard t-test * $p$ <0.05). We found the production of IL-18, an enterocyte derived cytokine induced by *Cryptosporidium* infection, [14,49] to be intact in the absence of TLR3 (Fig 5E, Standard t-test *** $p$ <0.001). Next, we measured IFNλ secretion from ileal punches of infected Tlr3$^{-/-}$ mice and wild type controls at day 2 of infection by ELISA. In the absence of TLR3, IFNλ production was reduced to the limit of detection (Fig 5F, Standard t-test *** $p$ <0.001). Note this reduction occurs despite an 8-fold higher infection in Tlr3$^{-/-}$. We conclude that the production of type III IFN during *Cryptosporidium* infection depends on TLR3 signaling.

## Discussion

We conducted a whole genome knock out screen to identify human genes that influence *Cryptosporidium* infection and host survival. We identified 35 genes with high confidence, and they implicate multiple pathways.

### Parasite attachment and invasion

The screen used an MOI of 3 in three consecutive rounds of selection. The high stringency may have enriched factors that act early in host-parasite interaction. Indeed, when comparing the level of infection of cell populations pre-screen and post-screen we noted differences as early as 3 hours post-infection (S3C Fig). Fittingly, five of the genes enriched in our screen encode steps in the synthesis of glycosaminoglycans. This provides further support for the notion that interactions between a parasite C-type lectin and host glycosaminoglycans are critical to parasite binding and invasion [50]. GPI anchor synthesis is also highly prominent among the enriched genes. GPI anchored proteins are preferentially targeted to the apical membrane of polarized cells [51], the membrane used by the parasite to invade. GPI anchored proteins are thus exposed to the parasite. Among them are glypicans which serve as the platform of apically displayed membrane associated glycosaminoglycans in the intestine [52]. The screen also identified the tetraspanin CD151. Interestingly, in infected cells this host protein is recruited to the host-parasite interface (S11 Fig). CD151 is critical to the uptake and intracellular trafficking of human cytomegalovirus and papillomavirus [53], and the related protein CD81 is required for the invasion of hepatocytes by *Plasmodium* sporozoites [54]. Tetraspanins act as scaffolds forming membrane microdomains that mediate adhesion, signaling, fusion and fission, and CD151 is well known for its role in integrin signaling [55]. Candidate FNDC3B contains a fibronectin type III domain involved in interactions with integrins and knockdown of this genes in HCT8 leads to a decrease in phosphorylation of PI3K [56]. Polymerization of host actin is a prominent feature of *Cryptosporidium* invasion and host modification and there is evidence for parasite engagement of host integrins and PI3K signaling [57,58] in this context.

### Cellular signaling and membrane trafficking

The screen also identified the kinase CSK, a negative regulator of Src family kinases. c-Src kinase was shown to play an important role in host actin polymerization during *Cryptosporidium* infection [23]. Traditionally, this has been viewed as aiding parasite infection; however, recent studies may suggest a more complex picture in which the cortical cytoskeleton might also act in host defense [24,59]. Src family kinases are also critical to pattern recognition receptor mediated detection of pathogens leading to the production of interferons and CSK is critical to tune this response [60,61].

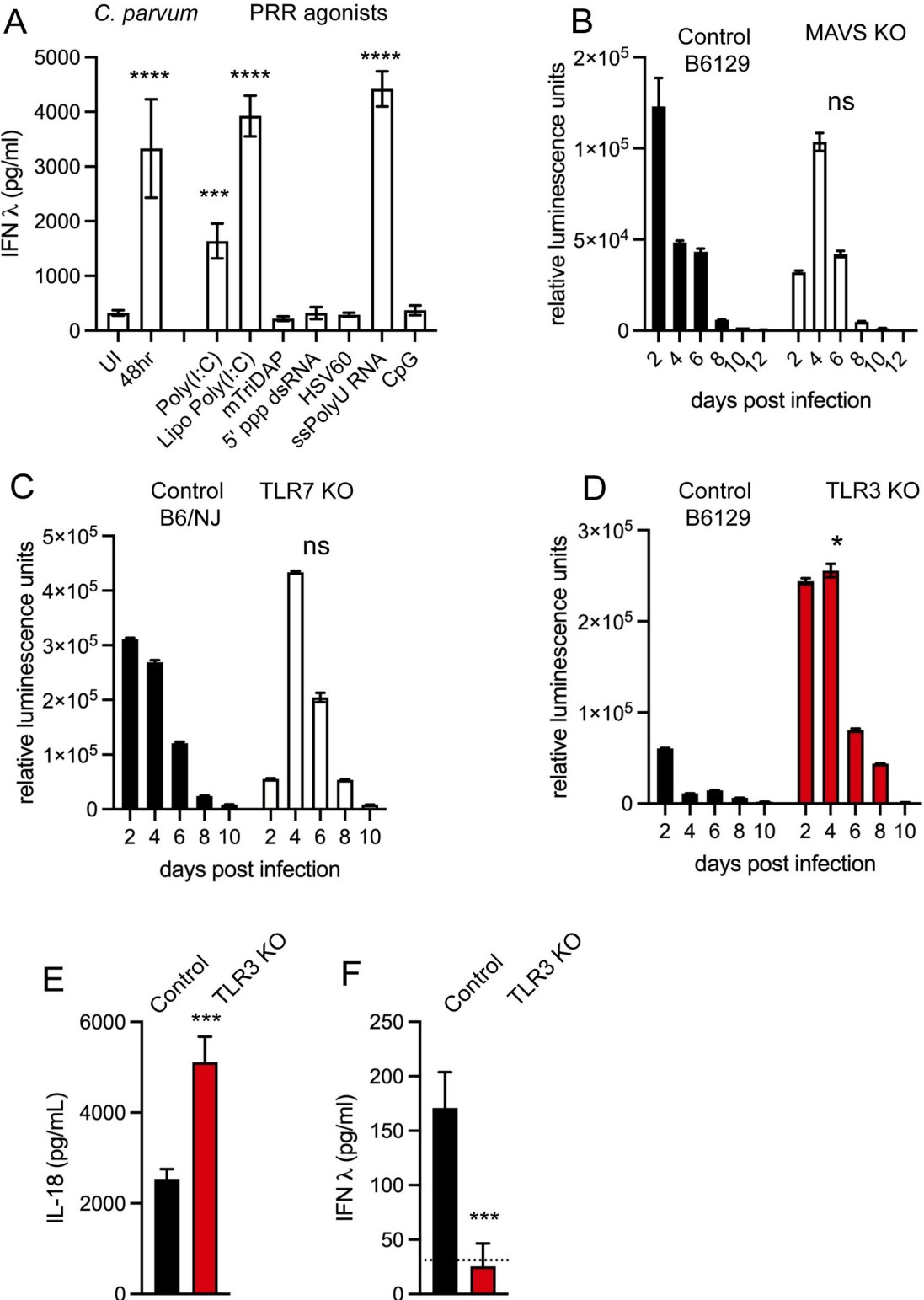

**Fig 5. TLR3 dependent recognition of *Cryptosporidium* infection.** We sought to identify the pattern recognition receptor that is activated to induce IFNλ production in response to *Cryptosporidium* infection. (A) ELISA of HCT-8 cultures infected with *C. parvum* or treated with the indicated agonist for 24 hours to assess IFNλ production in response to stimulus against a variety of pattern recognition receptors, *n* = 3. IFNλ is induced in response to the following stimuli: infection, Poly(I:C), lipofected Poly(I:C), and ssPolyU RNA One-way ANOVA with Dunnett's multiple comparisons *** *p* <0.001 **** *p* <0.0001. (B) Infection of wild

type control mice (B6129) compared to mice lacking MAVS. Fecal luminescence measured every 2 days. 3–4 mice per group. Representative of 2 biological replicates (1.18 and 1.04-fold increase) $n = 2$ Standard *t*-test of area under the curve across all biological replicates. (C) Infection of wild type control mice (C57B6N/J) compared to mice lacking TLR7. Fecal luminescence measured every 2 days. 3–4 mice per group. Representative of 3 biological replicates (1.3, 0.70, 2.2-fold increase) $n = 3$ Standard *t*-test of area under the curve across all biological replicates. (D) Infection of wild type control mice (B6129) compared to mice lacking TLR3. Fecal luminescence measured every 2 days. An increase of 8-fold in oocyst shedding was observed. 3–4 mice per group. Representative of 3 biological replicates (8, 12.4, 3.8-fold respectively) $n = 3$ Standard *t*-test of area under the curve across all biological replicates * $p < 0.05$. (E) IL-18 protein detected from ileal biopsies of infected mice wild type (B6129) compared to Tlr3$^{-/-}$ at day 2 post infection. Standard t-test *** $p < 0.001$. 11 mice per group, $n = 2$. (F) IFNλ protein detected from ileal biopsies of infected mice wild type (B6129) compared to Tlr3$^{-/-}$ at day 2 post infection. Standard t-test *** $p < 0.001$. 11 mice per group, $n = 2$. Dotted line represents the limit of detection.

Multiple hits may impact membranes and their trafficking including NBEAL2, TMEM30A, and RALGAPB. RALGAPB is an inhibitor of the small GTPases RalA and RalB, which in turn activates the exocyst complex. In epithelial cells the exocyst is critical to exocytosis as well as the dynamic remodeling of the actin cytoskeleton [62]. RalA activity is required for membrane recruitment to the *Salmonella typhimurium* infection site [63]. TMEM30A is an essential binding partner of P4 type ATPase flippases and directs the trafficking of the catalytic subunits from the trans-Golgi to the plasma membrane [64]. Deletion of TMEM30A leads to defects in both endocytosis and exocytosis due to a loss of the asymmetric distribution of phospholipids across the plasma membrane [65]. NBEAL2 is required for the formation of secretory granules in a variety of cells [66]. At least two hits act on nucleotide signaling, PDCL or phosducin-like G-protein, is a chaperone of G-protein beta gamma dimers [67] and erythrocyte G-proteins have been shown to play a role in *Plasmodium falciparum* invasion [68]. NPR3 is a G-protein coupled receptor which binds polypeptide hormones termed natriuretic peptides. Binding to this receptor inhibits adenylate cyclase and decreases cAMP [69].

## Innate immunity

By far the most prominent pathway to emerge from our screen was interferon signaling. This may initially appear counterintuitive as we found *in vivo* type III IFNs to limit parasite infection. However, as we demonstrate in this study, *Cryptosporidium* infection leads to the rapid accumulation of high levels of IFNλ in HCT-8 cultures, and we exposed cells to three rounds of heavy infection. The screen selected for host cell growth and survival, and we note that interferons are potent inducers of cell cycle arrest and cell death programs [70,71], and have been shown to induce apoptosis, necroptosis, and autophagy [72–75]. We directly tested this by treating HCT-8 with IFNλ, and found reduced cell viability (S4 Fig). This is consistent with the idea that the IFN signaling pathway emerged from the screen due to its role in host cell death which in vivo nonetheless acts as a mechanism to protect the host [76,77].

The role of IFNγ in cryptosporidiosis is well established, but there have also been reports of interferons directly produced by the enterocyte during *Cryptosporidium* infection. Barakat et al., described the production of type I interferons in response to *C. parvum* infection in a mouse cell line [13]. Transcriptional profiling of infected organoids from the lung and small intestine revealed a signature that was similarly interpreted as response to type I IFN signaling [15,16]. In contrast, Ferguson et al. recently reported type III interferon production in response to *C. parvum* infection in neonatal piglets and neonatal mice [12]. Their studies further suggest that IFNλ may block parasite invasion and promote barrier integrity during *C. parvum* infection.

In this study we show pronounced production of type III, but not type I interferons in human cells (Fig 6). We found this response to be initiated intrinsically in the infected cell (Fig 3H) and then amplified by an autocrine loop. Experiments *in vivo* demonstrated a protective role for IFNλ that did not require IFNγ or adaptive immunity but relied exclusively on signaling

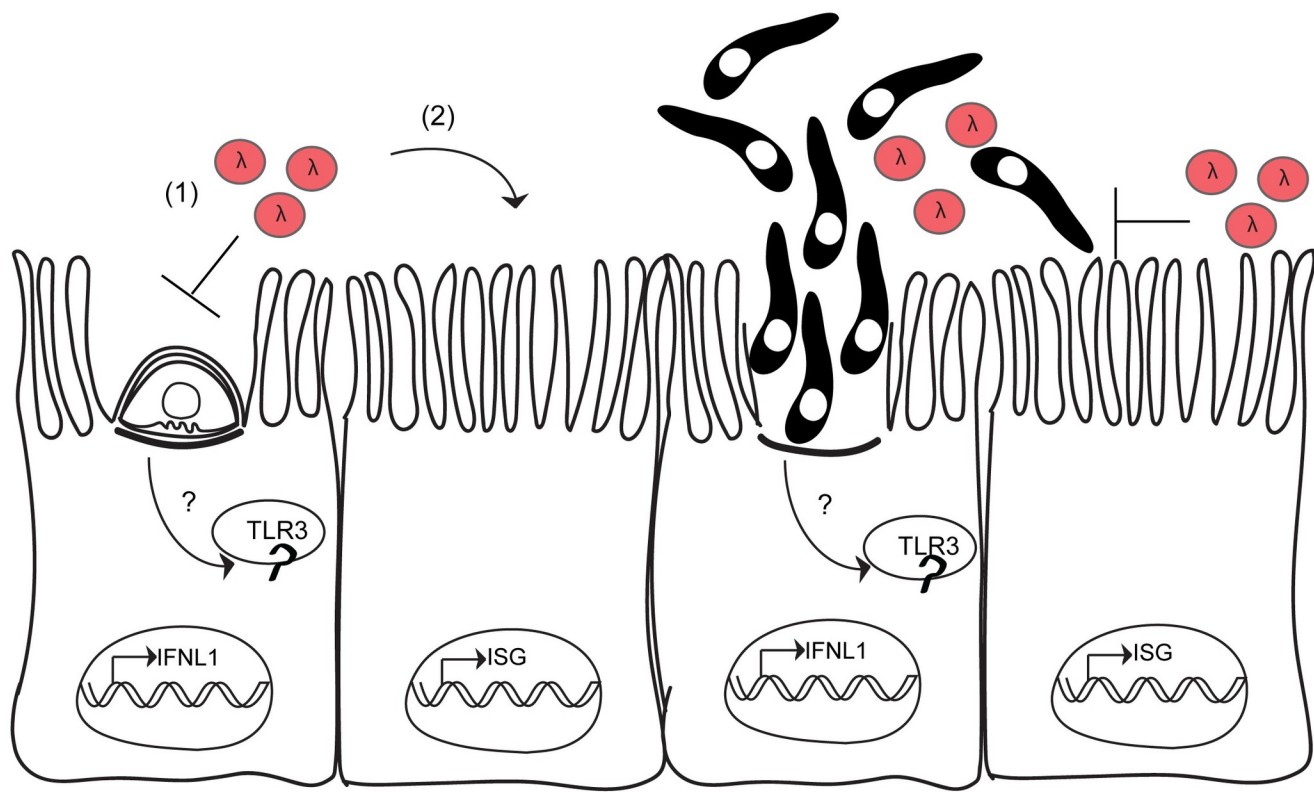

**Fig 6. Model.** *Cryptosporidium* infection leads to sensing by the endosomal pattern recognition receptor TLR3. Following activation of TLR3, IFNλ transcription is induced. Once IFNλ is secreted, it acts first on the infected cell (1), and then on uninfected bystanders (2) to induce the transcription of hundreds of ISGs. Lysis of the host cell by parasite egress releases intracellular contents, including IFNλ, further amplifying the type III interferon response. The protective effects of IFNλ in mice require intact STAT1 signaling in the intestinal epithelial cell lineage. One proposed mechanism of action for IFNλ is to block invasion of the parasite.

in enterocytes (Figs 4G and 5H). While our studies thus support epithelial cells as the primary producers of IFNλ we cannot exclude additional sources. Interestingly, we discovered the pattern recognition receptor, TLR3, to be required for type III interferon production. TLR3 recognizes dsRNA, and we used the synthetic analog, Poly (I:C), to induce IFNλ production. Previous studies have shown that injection of Poly (I:C) reduced *C. parvum* infection [78].

TLR3 is known to recognize other protozoan parasites including Neospora [79] and *Leishmania* [80] where it induces type I IFN production. For *Leishmania*, TLR3 recognition was dependent on the presence of a dsRNA virus found in certain parasite isolates. Interestingly, *Cryptosporidium* is also host to a dsRNA viral symbiont which could be a possible source of dsRNA recognized by TLR3 [81,82]. In contrast to leishmaniasis where type I IFN production exacerbates disease [83], the type III interferon response to *C. parvum* is host protective this is likely due to the different cell types, macrophages for *Leishmania* and intestinal epithelial cells for *Cryptosporidium*. There have also been reports of *Cryptosporidium* derived RNAs to be trafficked into the host cell nucleus during infection providing an additional potential trigger for TLR3 [84]. The route and mechanism by which such parasite derived RNA molecules would traffic into the host cell is unknown. While our screen did not identify TLR3 as a top candidate, UNC93B1, a protein critical to proper trafficking of endosomal TLRs [85], was a highly enriched gene. Cryptosporidiosis is most dangerous in children below the age of two years and it is thus noteworthy that TLR3 is poorly expressed in neonates, and that low levels of TLR3 in the intestinal epithelium have been linked to the heightened susceptibility of neonatal mice to rotavirus [86].

The increase in parasite burden in mice lacking TLR3 exceeded that of mice lacking IFNLR1 (8-fold compared to 2.5-fold), potentially pointing towards TLR3 functions beyond induction of type III interferon in epithelial cells. TLR3 is known to promote cross-priming of CD8+ T cells via DC phagocytosis of virus infected cells [87]. The absence of TLR3 signaling can impair the production of IL-12 by DCs [88] which is important for the control of *Cryptosporidium* infection [89]. The observation of decreased parasite burden in Ifnar1$^{-/-}$ mice is surprising but reproducible (Fig 4C) and may be due to an increase in effector T cell activation in the intestine which has been reported previously in these mice [90,91].

We recently reported that an enterocyte intrinsic NLRP6 inflammasome is activated by *Cryptosporidium* infection leading to release of IL-18 [14]. This IL-18 in conjunction with IL-12 then promotes downstream IFNγ production by ILCs [40]. Here we describe a second parasite detection mechanism that depends on TLR3 and produces a rapid IFNλ response that precedes the production of IFNγ by NK/ILCs and T cells. IFNλ has been shown to augment the IFNγ response of NK cells via a mechanism involving IL-12 [44]. However, loss of STAT1 exclusively in the enterocyte lineage led to almost a complete abrogation of the effects of IFNλ treatment (Fig 4H), arguing that the primary role of IFNλ is to act on the enterocyte. The protective effects of IFNλ treatment even in immunocompromised mice were striking. Treatment resulted in control in mice that are extremely susceptible to cryptosporidiosis [7,40] and lack central elements of innate and adaptive immunity. Treatment of immunocompromised individuals suffering from cryptosporidiosis remains very challenging [92,93]. Clinical trials for use of pegylated IFNλ have shown promise for the treatment of viral infections [94,95] and its efficacy against *Cryptosporidium* warrants further study.

## Materials and methods

### Ethics statement

All *in vivo* experiments were performed in accordance with protocols for animal care approved by the Institutional Animal Care and Use Committee at the University of Pennsylvania (#806292).

### Mice

C57/BL6J (stock no: 000664), B6129 (stock no: 101045), Ifng$^{-/-}$ (stock no: 002287), Ifnar$^{-/-}$ (stock no: 028288), Vil1 Cre (stock no: 021504), Mavs$^{-/-}$ (stock no: 008634), Tlr3$^{-/-}$ (stock no: 005217), Tlr7$^{-/-}$ (stock no: 008380) mice were purchased from Jackson Laboratories. C57/BL6 (Model no: B6NTac), Rag2$^{-/-}$ (Model no: RAGN12), Rag2$^{-/-}$ Il2rg$^{-/-}$ (Model no: 4111) mice were purchased from Taconic Biosciences. Vil1-Cre (stock no: 021504) were purchased and STAT1flox mice were generated as previously described [96] and maintained in house. Ifnlr1$^{-/-}$ mice [97] (Bristol Meyers Squibb) were maintained in house. Mice used in this study were males or females between 6–10 weeks of age. Mice were neither co-housed nor were littermate controls used unless explicitly stated. Initial experiments showed that *Cryptosporidium* susceptibility to type III interferon was unchanged whether mice were confused prior to infection or not (S6 Fig). All mice were sex and age matched for each experiment. No differences in infection were observed between male and female mice.

### Cells, parasites, and infections

HCT-8 cells (ATCC) were maintained in RPMI supplemented with 10% FBS at 37˚C and 5% CO$_2$. Wild-type *Cryptosporidium parvum* oocysts used in this study were purchased from Bunchgrass Farms (Dreary, ID). Parasites expressing Tandem mNeon were generated in a

previous study [25]. For *in vitro* infections oocysts were incubated in a (1:3) bleach: water solution for 10 minutes at 4˚C, centrifuged then resuspended in a 0.08% solution of sodium deoxytaurocholate and incubated at 16˚C for 10 minutes. Oocysts were then washed in PBS and finally resuspended in infection media (complete RPMI with 1% FBS) and added directly to host cells.

*C. parvum* oocysts used for all *in vivo* experiments are mouse adapted mCherry and Nanoluciferase expressing [14]. Mice were infected with 50,000 *C. parvum* oocysts by oral gavage unless otherwise noted.

## Killing assay

6 well cultures grown to 60% confluency were infected with $5 \times 10^5$, $1.25 \times 10^6$, $2.5 \times 10^6$, $3.75 \times 10^6$, or $5 \times 10^6$ oocysts per well in biological duplicate. Following a 72-hour infection, cells were trypsinized and incubated in a 1:4 solution of Trypan Blue. Cells were counted and Trypan Blue exclusion used to determine viability.

## CRISPR screen

The Brunello CRISPR sgRNA library [31] was optimized for on-target activity and to minimize off-target effects. Brunello contains four sgRNAs per protein coding gene in the human genome. HCT-8 in media containing 1μg/mL polybrene were spinfected (2 hours, 30˚C at 1,000xg) with lentivirus to produce a constitutively expressing Cas9 cell line (lentiCas9-Blast, plasmid#52962, addgene). Following a 7-day selection with blasticidin (1.5μg/mL), cells were diluted to generate clonal Cas9 expressing cell lines. To measure Cas9 activity these cells were subjected to an EGFP reporter assay for Cas9 activity. Cas9 expressing cells were spinfected with lentiXPR_011, encoding an EGFP and a sgRNA targeting EGFP. 24 hours post spinfection cells were flow sorted to assess green fluorescence. Cells lacking or stably expressing EGFP expressing were used as controls for flow cytometry.

We sought to achieve 1000-fold coverage across multiple biological replicates of the screen. Each replicate achieved 500-fold coverage. Per mL of the Brunello library there were $4.2 \times 10^7$ lentiviruses/guides. We infected at an MOI of 0.4 therefore $1.02 \times 10^8$ cells were transduced with the library. These cells were trypsinized and spinfected as before into a total of six 6-well plates. Plates were incubated at 37˚C, 5% CO2 for 24 hours then media was changed to include 1ug/mL puromycin for selection of transduced cells. Seven days later, cells were trypsinized and expanded into 12 T-175 flasks. After expansion, genomic DNA was isolated from four flasks as the input population and at least $4 \times 10^7$ cells (500-fold coverage) were passaged into 4 T-175. These 4 T-175 were then infected with a 90% kill dose of *C. parvum* oocysts. After 72 hours media was replaced with fresh media and cells were allowed to recover. Once confluent, cells were trypsinized and seeded into new flasks to be re-infected while at least $4 \times 10^7$ cells were taken for genomic DNA extraction. In total, the population was subjected to three rounds of successive *C. parvum* infection.

Genomic DNA was extracted using the QIAamp DNA Blood Maxi kit (Qiagen). sgRNAs were amplified by PCR as described [31]. Read counts were normalized to reads per million in each condition.

## MAGeCK analysis of CRISPR screen

Data from our CRISPR screen was analyzed using MAGeCKFlute in R [98]. MAGeCK uses a negative binomial to test for differences in sgRNA abundance between conditions [32]. The input population for Clone K was compared to the output of each round of infection. For Clones C and I input was compared only to the final population. Results shown are for the

combined data of screens with Clone I and K. Clone C was excluded due to poor sequencing depth. Genes with three or four sgRNAs positively ranked by the robust ranking aggregation (RRA) algorithm and an FDR of less than 0.05 were considered significantly enriched. Pathways identified by GSEA that included multiple genes of the top candidates were displayed in Fig 1.

## RNAi screen

siRNAs targeting top screening candidates were purchased from Ambion (ThermoFisher Scientific, Waltham, MA). Both scrambled non-targeting siRNAs and a positive transfection control RNA targeting GAPDH were included. siRNAs were delivered to 96 wells at 50% confluency using Lipofectamine RNAiMax (ThermoFisher Scientific, Waltham, MA) to a final concentration of 100nM per well. 24 hours later, wells were infected with 25,000 *C. parvum* oocysts. At 48 hours post infection, cells were lysed and RNA extracted using the Rneasy Mini Kit (Qiagen). Knockdown of target genes was assessed by qPCR. Host cell viability was measured by MTT (3-(4,5-dimethylthiazol-2-yl)-2,5-diphenyltetrazolium bromide) assay. Briefly, media was removed from all wells and replaced with 100μL of fresh RPMI. 10μL of 12mM MTT solution was added to each well. Plates were incubated at 37˚C for 4 hours. Then all media was removed and replaced with 50μL of DMSO (Sigma, St. Louis, MO) and mixed thoroughly by pipetting up and down. Following a 10 minute incubation at 37˚C, plates were read for absorbance at 540nm.

## Immunofluorescence assay

Infected HCT-8 coverslip cultures were fixed in 4% paraformaldehyde and permeabilized with 0.1% Triton X-100 for 10 minutes each at room temperature. Samples were blocked in 3% Bovine Serum Albumin (BSA) for 1 hour and primary antibodies were diluted in 3% BSA. Anti-CD151 (ab33315, Abcam) was diluted 1:100 and anti-Tryptophan synthase beta [41] was diluted 1:1000. Secondary antibodies (ThermoFisher) were diluted 1:1000 in 3% BSA. FITC conjugated phalloidin (F432, ThermoFisher) was included in the secondary antibody incubation. Cell nuclei were labeled with Hoechst 1:10,000 for 5 minutes and coverslips were mounted using Vectashield (Vector Laboratories). Slides were imaged using a Leica DM6000 Widefield microscope.

## RNA sequencing

Total RNA was extracted using the RNeasy Mini (48-hour) RNeasy Micro (10-hour) kit (Qiagen). cDNA was synthesized using the SMART-Seq v4 Ultra Low Input RNA Kit (Takara Bio USA), and barcoded libraries were prepared using the Nextera XT DNA Library Preparation Kit (Illumina). Total RNA and libraries were quality checked and quantified on an Agilent Tapestation 4200 (Agilent Technologies). Samples were pooled, and single-end reads were run on a NextSeq 500 (Illumina).

Reads were pseudo-aligned to the Ensembl *Homo sapiens* reference transcriptome v86 using kallisto v0.44.0 [99]. At least 85% of reads aligned to the human genome in all samples with 3% or less mapping to the *Cryptosporidium parvum* genome. In R, transcripts were collapsed to genes using Bioconductor tximport [100] and differentially expressed genes were identified using Limma-Voom [101,102]. Gene set enrichment analysis (GSEA) was performed using the GSEA software and the annotated gene sets of the Molecular Signatures Database (MSigDB) [103].

## qPCR

RNA concentrations were measured by NanoDrop (ND-1000; Thermo Fisher Scientific, Waltham, MA) for each sample and an equal amount of cDNA was prepared using SuperScript IV Reverse Transcriptase (Thermo Fisher Scientific, Waltham, MA). Following reverse transcription, a 20μL reaction was loaded into a ViiA 7 Real Time PCR system (Thermo Fisher Scientific, Waltham, MA). The following conditions were used: Initial incubation 3 min at 95˚C, 40 cycles of 95˚C for 15 sec and 60˚C for 30 sec. A single melt curve and ΔΔCt method was used to determine relative expression with GAPDH used as the housekeeping gene. See S1 Table for list of primers.

## Western blot

24 well HCT-8 cultures grown to 60% confluency were infected with $2x10^5$ C. parvum oocysts in RPMI containing 1% serum for the indicated time. Media was removed and cells were lysed in Pierce IP Lysis Buffer (ThermoFisher Scientific, Waltham, MA), supplemented 1:100 with protease inhibitor cocktail (Sigma St. Louis, MO). Lysates were incubated on ice 10 minutes, then spun at 20,000 g for 10 min at 4˚C. The cleared lysate was removed and flash frozen. Cleared lysates were thawed on ice and protein concentration was assessed by BCA (23225, ThermoFisher Scientific, Waltham, MA). 18μg of sample was loaded per well diluted 1:1 with freshly prepared 2X Laemmli Sample buffer (BioRad Hercules, CA) + β-Mercaptoethanol (1:20) (Sigma St. Louis, MO) and boiled for 10 minutes at 95˚C. 20 μL sample was loaded per each lane of an any KD Mini-PROTEAN TGX Precast Protein Gel (BioRad Hercules, CA) and run at 150 V for 1 hour. Wet transfer to a 0.45 μm pore size pre-cut Nitrocellulose membrane (ThermoFisher Scientific Waltham, MA) was conducted at 20V for 2.5 hours at room temperature. The Nitrocellulose membrane was blocked for 1 hour at room temperature using Intercept (TBS) Protein-Free Blocking Buffer (LI-COR Lincoln, NE). Primary antibody was incubated at room temperature for 2 hours in Intercept(TBS) Protein-Free Blocking Buffer with 0.01% Tween20 (Sigma, St. Louis, MO) using STAT1 1:1000 (#14994, Cell Signaling Technology), phosphor STAT1 Y701 1:1000 (ab29045, Abcam) and alpha-tubulin 1:5000 (ab7291, Abcam). The membrane was washed 3 times with PBS with 0.01% Tween20 (Sigma, St. Louis, MO). Secondary antibody was incubated at room temperature protected from light for 1 hour in Intercept (TBS) Protein-Free Blocking Buffer with 0.01% Tween20 (Sigma, St. Louis, MO) using IRDye 800CW Goat anti-Mouse IgG secondary antibody at 1:10,000 (LI-COR Lincoln, NE) and IRDye 680RD Goat anti-Rabbit IgG secondary antibody at 1:10,000 (LI-COR, Lincoln, NE). After 3 PBS + 0.01% Tween20 (Sigma, St. Louis, MO) washes, the membrane was imaged on the Odyssey Infrared Imaging System v3.0 (LICOR, Lincoln, NE).

## ELISA

96 well HCT-8 cultures grown to 60% confluency were infected with 25,000 *C. parvum* oocysts. At the indicated timepoint post infection, supernatants were removed and spun at 1,000xg for 10 minutes to pellet debris. Supernatants were frozen at -80˚C. IFNβ and IFNλ protein levels from HCT-8 cultures were measured by Human IFN-beta DuoSet ELISA (DY814, R&D Systems) and Human IL29/IL28B (IFN-lambda 1/3) DuoSet ELISA (DY1598B, R&D Systems). Protein levels of IFNλ from intestinal biopsies were measured by Mouse IL28B/IFN-lambda 3 DuoSet ELISA (DY1789B, R&D Systems). Protein levels of IL-18 from intestinal biopsies were measured by ELISA (BMS618-3, ThermoFisher, Waltham, MA). Assays were performed according to the manufacturer's instructions.

## Flow sorting of infected cells

HCT-8 6 well cultures were infected with $1x10^6$ *C. parvum* Neon oocysts. 10 hours later, cells were trypsinized in TrypLE (ThermoFisher), washed with PBS, and passed through a 40 μm filter. Cells were sorted using a BD FACSJazz Sorter (BD Biosciences). Uninfected HCT-8 were used to gate on singlets. 10,000 positive cells and 10,000 negative cells were sorted from three independent biological replicates directly into RLT Lysis buffer (Qiagen).

## Ileal biopsies

Three 5mm punch biopsies were taken from the distal small intestine of each mouse. Punches were incubated in complete RPMI for 18 hours. Supernatants were then used for ELISA.

For qPCR, punches were placed in RNAlater (Sigma) at 4˚C until RNA was extracted using the RNeasy Mini Kit (Qiagen)

## Nanoluciferase assay

To monitor infection *in vivo*, 20mg of fecal material was resuspended in 1mL of lysis buffer. Samples were shaken with glass beads for 5min at 2,000 rpm. Samples were briefly centrifuged to pellet any floating material and the cleared lysate was mixed 1:1 with prepared Nanoluciferase solution (substrate: lysis buffer 1:50). Luminescence was measured using a Promega Glo-Max plate reader.

## Cytokine neutralization and administration

To neutralize IFNλ, 20μg of Anti IL28A/B (Clone 244716, MAB17892, R&D Systems, Minneapolis, MN) was infected intraperitoneally one day prior and each day following infection for the duration.

For administration of Ifnλ2 (250–33, Peprotech, Cranbury, NJ), 1μg, unless otherwise noted, was injected intraperitoneally daily beginning at 6–8 hours prior to infection and then each day of the infection. For administration of Ifnβ (8234-MB-010, R&D Systems, Minneapolis, MN) 1μg was injected intraperitoneally daily beginning at 6–8 hours prior to infection and then each day of the infection.

## Histology

Tissue from the lower third of the small intestine was flushed with 10% neutral buffered formalin (Sigma, St Louis, MO, USA), then 'swiss-rolled' and fixed overnight. Fixed samples were paraffin-embedded, sectioned, and stained with hematoxylin and eosin for detailed histologic evaluation. Slides were evaluated by a board-certified veterinary pathologist in a blinded fashion for quantitative measurements of number of parasites, villus/crypt architectural features, and semi-quantitative scores for villus epithelium lesions as previously described [42].

## PRR agonist screen

Agonists of pattern recognition receptors were purchased from Invivogen. Cells were seeded into 96 well plates and at 60% confluency, cells were either infected with C. parvum (25,000 oocysts per well) or treated with an agonist. 10μg/mL LMW Poly (I:C) was either lipofected or added to the medium. The following agonists were delivered with Lipofectamine: 5'ppp RNA (10μg/mL), mTriDAP (10μg/mL), HSV60 (5μg/mL), ssPolyU RNA (10μg/mL), and CpG ODN (5μM). After 24 hours the media was removed for ELISA and the cells were lysed and RNA extracted (RNeasy Mini Kit, Qiagen).

## Statistical methods

Mean +/- SD are reported. For *in vivo* experiments, fold change of area under the curve across all replicates was used to determine statistical significance. When measuring the difference between two populations, a standard *t*-test was used. For datasets with 3 or more experimental groups, a one-way ANOVA with multiple comparison's test was used. For datasets with 2 or more experimental groups and an additional factor of time, a two-way ANOVA with multiple comparison's test was used. Simple linear regression was used to determine the goodness of fit curve for host cell killing by *C. parvum*. *P* values of less than 0.05 were considered significant. These tests were performed in GraphPad Prism or in R. Additional replicates of *in vivo* data can be found in S9 and S10 Figs.

## Supporting information

**S1 Fig. Repeated infection of HCT-8 alone does not lead to increased host cell viability.** Cas9 expressing HCT-8 cells were infected successively with a 90% kill dose (MOI = 3) but in contrast to the screen this was done in the absence of the sgRNA library. Host cell viability was assessed by Trypan Blue exclusion and no change in susceptibility was observed under these conditions.
(TIF)

**S2 Fig. Impact of siRNA knockdown of screening hits on host cell survival upon *C. parvum* infection.** We used siRNA treatment to knockdown transcripts of genes identified in our screen and assessed host cell viability following infection. (A) Relative expression of genes targeted for knockdown normalized to the scrambled (scr) siRNA control. *n* = 2. (B) Knockdown of top candidates does not affect host cell viability in the absence of infection. MTT assay normalized to uninfected scrambled (scr) siRNA control. *n* = 2. (C) Knockdown of candidates leads to an increase in host cell viability during *C. parvum* infection. MTT assay normalized to uninfected scrambled (scr) siRNA control. *n* = 2.
(TIF)

**S3 Fig. Cells emerging from the screen are less susceptible to infection with *C. parvum*.** The population of Cas9-Clone K expressing cells transduced with the library prior to the screen were compared to the population following the last round of selection. (A) The pre- and post-screen populations were infected with Nanoluciferase expressing parasites for 48 hours. Parasite growth was measured by Nanoluciferase assay. Standard *t*-test $p < 0.001$ **** *n* = 3. (B) Cells were infected with *C. parvum* for 72 hours. Host cell viability was assessed by Trypan Blue exclusion. *n* = 3 Standard *t*-test. (C) The pre- and post-screen populations were infected with Nanoluciferase expressing parasites for the indicated time points. Parasite growth was measured by Nanoluciferase assay. Two-way ANOVA $p < 0.001$ **** *n* = 3.
(TIF)

**S4 Fig. Type I and type III interferons in HCT-8.** (A) GSEA plot showing Interferon Alpha Beta Signaling and Interferon Lambda Response signatures identified at 48 hours post infection. Closed circles represent genes that make up the core enrichment of the signature. Note that many of the genes overlap. Alpha Beta: Net enrichment score = 2.9, p-value <0.0001, Lambda: Net enrichment score = 3.04, *p* value <0.0001. (B) Protein levels of IFNβ and IFNλ following lipofection with 10μg/mL Poly(I:C) as measured by ELISA. Note the maximal production of IFNβ is 27-fold higher than that observed during *C. parvum* infection. (C) MTT assay normalized to untreated, uninfected control. HCT-8 were infected for 48 hours following 16 hours treatment with IFN at the indicated doses. One-way ANOVA with Dunnett's

multiple comparisons test * $p < 0.05$ **** $p < 0.0001$.
(TIF)

**S5 Fig. Comparison of uninfected bystander cells in infected cultures to cells from uninfected cultures.** Volcano plot of RNA-seq of Neon negative bystander cells (see Fig 3H) were compared to uninfected controls. Only a relatively small number of differentially expressed genes was observed and no enrichment of an IFN signature was noted in either dataset (see S5 Table for GSEA results).
(TIF)

**S6 Fig. Co-housing of Ifnlr1$^{-/-}$ mice.** 4-week-old mice were co-housed for 3 weeks prior to infection. Fecal luminescence measured every two days in C57/BL6 wild type mice and mice lacking the type III interferon receptor Ifnlr1$^{-/-}$ following infection with 50,000 *C. parvum*. Aggregate of 2 biological replicates is shown. An average 2-fold increase was observed across 2 biological replicates. 4 mice per group.
(TIF)

**S7 Fig. BL6 and Ifnlr1 KO mice exhibit similar baseline and post-infection intestinal pathology scores.** (A) Histology scoring from uninfected BL6 wild type mice and mice lacking the type III interferon receptor Ifnlr1$^{-/-}$. No differences were observed. (B) Hematoxylin and eosin stained sections of the distal small intestine of BL6 and IFNLR1 KO mice prior to and 4 days post infection. (C) Histology scoring from BL6 wild type mice and mice lacking the type III interferon receptor Ifnlr1$^{-/-}$ infected with 50,000 *C. parvum* and uninfected controls. One-way ANOVA with Šídák's multiple comparisons test * $p < 0.05$ ** $p < 0.01$ *** $p < 0.001$ **** $p < 0.0001$. Differences between uninfected Ifnlr1$^{-/-}$ mice compared to infection tended to be more statistically significant than those observed between uninfected and infected BL6 mice.
(TIF)

**S8 Fig. IFNβ treatment of Ifng$^{-/-}$ mice.** Ifng$^{-/-}$ mice were injected i.p. with 1μg Ifnβ daily on days 0–3 of infection. Mice were infected with 20,000 *C. parvum* oocysts. The total area under the curve of fecal luminescence for the 3-day infection is shown. Two biological replicates are shown resulting in a decrease of 2.9 and 1.8-fold. Standard *t*-test of area under the curve across two biological replicates * $p < 0.05$.
(TIF)

**S9 Fig. Biological replicates of infections shown in Fig 4.** Fecal luminescence was measured every two days following infection with 50,000 *C. parvum*. (A) BL6 wild type mice, Ifnar1$^{-/-}$, and Ifnlr1$^{-/-}$ 4 mice per group, 2 additional biological replicates shown. (B) C57/BL6 mice were treated with anti-Ifnλ2/3 antibody or an isotype control daily via intraperitoneal (i.p.) injection and infected. One additional biological replicate shown. (C) Ifng$^{-/-}$ mice were injected i.p. with indicated doses of Ifnλ2 daily for days 0–3 of infection. Mice were infected with 20,000 *C. parvum* oocysts. The total area under the curve of fecal luminescence for the 3-day infection is shown. 2 mice per dose. One additional biological replicate shown. (D) Ifng$^{-/-}$ mice were injected i.p. with 1μg of Ifnλ2 beginning at day 0 and each day for the duration of the infection. Mice were infected with 20,000 *C. parvum* oocysts. One additional biological replicate shown. (E) Wild type mice (B6), mice lacking T cells (Rag2$^{-/-}$), and mice lacking NK cells, ILCs, and T cells (Rag2/Il2rg$^{-/-}$) were treated with 1μg of Ifnλ2 daily for the days 0–3 of infection. The total area under the curve of fecal luminescence for the 3-day infection is shown. Two additional biological replicates shown. (F) Villin Cre STAT1 flox mice or littermate Cre negative controls were treated with 1μg of Ifnλ2 daily for the days 0–3 of infection. The total area under the curve of fecal luminescence for the 3-day infection is shown. One additional biological

replicate shown.
(TIF)

**S10 Fig. Biological replicates of Fig 5.** Fecal luminescence measured every two days following infection with 50,000 *C. parvum*. (A) Infection of wild type control mice (B6129) compared to mice lacking MAVS. One additional biological replicate is shown. (A) Infection of wild type control mice (C57B6N/J) compared to mice lacking TLR7. Two additional biological replicates shown. (C) Infection of wild type control mice (B6129) compared to mice lacking TLR3. Fecal luminescence measured every 2 days. Two additional biological replicates shown.
(TIF)

**S11 Fig. CD151 localizes to the *Cryptosporidium* invasion site.** Immunofluorescence of HCT-8 infected with *C. parvum* at 1 hour post infection. *C. parvum* (red), CD151 (gray), actin (green) Hoechst label nuclei. Scale bar 10μm in top panel, scale bar 5μm in bottom panel.
(TIF)

**S1 Table. List of primers.**
(XLSX)

**S2 Table. siRNA sequences.**
(XLSX)

**S3 Table. CRISPR screening data.** Summary read count file used for analysis. Summary of MAGeCK analysis. GSEA for screening data.
(XLSX)

**S4 Table. Differential gene expression and GSEA for RNA-seq at 48 hours post infection.**
(XLSX)

**S5 Table. Differential gene expression and GSEA for RNA-seq at 10 hours post infection.**
(XLSX)

## Author Contributions

**Conceptualization:** Alexis R. Gibson, Adam Sateriale, Jennifer E. Dumaine, John G. Doench, Christopher A. Hunter, Boris Striepen.

**Data curation:** Alexis R. Gibson, Adam Sateriale, Daniel P. Beiting.

**Formal analysis:** Alexis R. Gibson, Julie B. Engiles.

**Funding acquisition:** Christopher A. Hunter, Boris Striepen.

**Investigation:** Alexis R. Gibson, Adam Sateriale, Jennifer E. Dumaine, Ryan D. Pardy, Jodi A. Gullicksrud, Keenan M. O'Dea.

**Methodology:** Alexis R. Gibson, Julie B. Engiles, Jodi A. Gullicksrud.

**Resources:** John G. Doench, Daniel P. Beiting.

**Software:** Daniel P. Beiting.

**Supervision:** Christopher A. Hunter, Boris Striepen.

**Validation:** Alexis R. Gibson.

**Visualization:** Alexis R. Gibson, Ryan D. Pardy, Keenan M. O'Dea.

**Writing – original draft:** Alexis R. Gibson, Boris Striepen.

**Writing – review & editing:** Alexis R. Gibson, Christopher A. Hunter.

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
