## [Decision Letter · Decision Letter 0]

2 Nov 2021

Dear Dr Striepen,

Thank you very much for submitting your manuscript “A genetic screen identifies a protective type III interferon response to Cryptosporidium that requires TLR3 dependent recognition “ for review by Plos Pathogens. Your article has been reviewed by our editors and peer reviewers. The reviewers appreciated the attention to an important topic, but raised some substantial concerns about the manuscript as it currently stands. Additional experiments are recommended as well as comprehensive changes how the data is presented. These issues must be addressed before we would be willing to consider a revised version of your study.

We cannot make any decision about publication until we have seen the revised manuscript and your response to the reviewers' comments. Your revised manuscript is also likely to be sent to reviewers for further evaluation.

Sincerely,

Philipp Olias

Associate Editor

PLOS Pathogens

Vern Carruthers

Section Editor

PLOS Pathogens

Kasturi Haldar

Editor-in-Chief

PLOS Pathogens

orcid.org/0000-0001-5065-158X

Michael Malim

Editor-in-Chief

PLOS Pathogens

orcid.org/0000-0002-7699-2064

Reviewer's Responses to Questions

**Part I - Summary**

Reviewer #1: The manuscript by Gibson AR et al. examined the role of type III interferon (IFN-λ) in Crytosporidium infection. The authors performed a cell-death-based genome-wide CRISPR knockout screen to identify host factors that support Crytosporidium infection and found several genes in the IFN signaling pathway. The authors found that IFN-λ was induced during Crytosporidium infection in a replication-dependent manner in vitro and mediated via TLR3 in vivo. Crytosporidium fecal shedding was enhanced in mice lacking IFN-λ receptors. A protective role of IFN-λ in Crytosporidium infection has been previously reported (Ferguson SH et al., Cell Mol Gastroenterol Hepatol, 2019), limiting the novelty of the study. There also seems to be a disconnection between the in vitro screening data (IFN-λ signaling responsible for Crytosporidium induced cytotoxicity) and suppression of Crytosporidium infection in vivo. However, the overall manuscript is well-organized and clear, the conclusions are valid, and most experiments are well-controlled. I have a few suggestions and comments to help improve the study.

Reviewer #2: In this paper a large amount of experimental work was performed to show the importance of interferon signaling in regulating Cryptosporidium infection. First, a genome wide screen was performed to identify host factors that impact Cryptosporidium infection. The screen appears to have been carried out in a robust manner and with a reasonable coverage (500). Three biological replicates were carried out with three independent Cas9-expressing clones.

Secondly, RNAseq revealed that interferon signaling is induced by Cryptosporidium infection. qPCR and ELISA showed that specifically IFN type III signaling is induced. The authors next turned to in vivo models and made use of several available knock out mouse strains to investigate the impact of interferon signalling on Cryptosporidium infection.

I enjoyed reading this work and find it to be an important contribution to the community. The connection between the genome wide screening part of the work and the rest of the story is not always clear – it felt somewhat like reading two different projects lumped together. While the screen is undoubtedly interesting, I felt that the data was not presented in an optimal or transparent manner, and more importantly, an in-depth validation / further characterization of the identified genes was not carried out. I have listed several questions, comments and suggestions. Most importantly, comment 5 and 6 – I’d love to know more about how loss of particular genes / pathways impact Cryptosporidium development.

Reviewer #3: In this paper, authors used an unbiased screening technique to identify genes determining human host cell death following infection with Cryptosporidium parvum. They identified 35 genes significantly associated with cell death, belonging to 4 groups: “IFN signaling”, “GPI anchors”, “glycosaminoglycan biosynthesis” and “miscellaneous”. Following this screen, the authors explored more in depth the role of IFNs, and more precisely type 3 IFNs (IFNλ1,2 and 3) in the pathogenesis of cryptosporidiosis. Using both in vitro and in vivo models, they describe the protective effect of type 3 IFNs during cryptosporidiosis. They show that the source of IFNλ are infected cells through the activation of a TLR3 dependent pathway. IFNλ secreted by infected cells would later stimulate the infected cell and bystander cells in a paracrine fashion, leading to the expression of IFNλ-dependent ISGs. The genes identified by screening belonging to other groups than the “IFN signaling” group have not been investigated more.

However, the description of type 3 IFNs as an important part of the innate immune response to Cryptosporidium infection at the intestinal epithelium is not new, as this role had already been evidenced in an earlier paper (Ferguson et al., 2019) using in vitro (non-transformed porcine jejunal epithelial cells) and in vivo (neonatal C57BL/6 mice) models. The current findings extend the role of type III IFN to human cells in vitro.

Reviewer #4: Cryptosporidium is a major cause of diarrheal disease that targets intestinal epithelial cells, but many open questions remain about its interaction with the host. The study by Gibson et al. undertook a genome wide knockout screen in a human epithelial cell line and identified known and novel host factors that promote cell survival. The most significant set of genes belonged to the type III interferon (IFNL) signaling pathway, which is more commonly associated with antiviral immunity than antiparasitic immunity. The authors go on to show that IFNL is preferentially stimulated by Cryptosporidium infection, requires live parasites, and correlates with parasite burden. Studies with mouse-adapted Cryptosporidium show that the IFNL response is both necessary and sufficient for optimal control of parasite shedding. Additionally, IEC-intrinsic expression of the downstream STAT1 transcription factor is important for parasite control and IFNL therapeutic efficacy. Together, these studies provide an important resource genetic screen for host genes that control infection by this understudied pathogen and perform mechanistic studies that characterize the involvement of IFNL pathway in anti-parasitic immunity in vivo. This is an important paper and findings will be of significant interest to the fields if Cryptosporidium and intestinal immunity. Conclusions are generally well-supported by convincing data, and I have only a few specific points of critique.

**Part II – Major Issues: Key Experiments Required for Acceptance**

Reviewer #1: Major points:

(1) As mentioned above, there is a significant lack of connection between the CRISPR screen and IFN-λ inhibiting Crytosporidium infection in mice. For instance, does knocking out STAT1 (#1 hit) prevent Crytosporidium induced cell death in HCT-8 cells? How does that reconcile with increased Crytosporidium infection in vivo (Fig. 4H)? This needs to be discussed in the paper.

(2) I have several problems with Fig. 4. Panel (A): IFN-λ is generally rapidly induced post infection. 12 hr or at least day 1 data would be revealing (higher fold induction than day 2). Panel (B): huge error data. More replicates needed. Panel (C): several virus infection studies have shown a redundant role of type I IFN and IFN-λ (Crotta S et al., PLOS Pathog, 2013; Lin JD et al., PLOS Pathog, 2016). The increase of Crytosporidium in IL28RA knockout mice was far less than that in the STAT1 floxed, Villin-Cre mice (Fig. 4H). The authors need to test Crytosporidium infection in Ifnar1, Ifnlr1 double knockout mice to show that IFN-λ alone controls infection. Panel (D and F): anti-IFN-β and recombinant IFN-β need to be included as controls to show specificity.

(3) The protective role of IFN-λ in Crytosporidium infection has been reported. The cell types responsible for IFN-λ production and the upstream sensors are potentially novel and unfortunately not examined in depth. What cells (intestinal epithelial cells, intraepithelial lymphocytes, lamina propria hematopoietic cells) are the major producer of IFN-λ in response to Crytosporidium infection? The authors showed a role of TLR3 in IFN-λ expression. However, in what compartment? Do TLR3 floxed, Villin-Cre mice show the same phenotype as full TLR3 KO mice?

Reviewer #2: 1. What happens when you perform 3 rounds of 90% killing dose on non library-transduced cells? Is this dose sufficient to kill all host cells, or do you select for spontaneously resistant cells? If yes, it might be interesting to test expression of interferon stimulated genes in those cells. Please comment on this.

2. Figure 1A: at the dose where you obtain 90% killing, are all your cells infected with the parasite? How can you exclude the influence of death of bystander cells? Please show infection rate in addition to cell viability data. Did you perform technical and biological replicates (different batches of Crypto) in the infection assay? Can you make it a bit clearer what MOI you used?

3. Genome wide screening data is shown from clone K (Figure 1D) and clones K+I (Figure 1E). What about the data from clone C? Why was clone F excluded, seeing as this has the highest level of Cas9 activity? Please illustrate how well the hits from your three replicates overlap with each other.

4. It seems to me that 3 rounds of 90% killing dose is a very stringent selection, and that it’s possible (likely?) you enriched for genes that are essential for early steps in parasite attachment and invasion and lost hits that may be essential for parasite development. I think this is worth commenting on.

5. After each round of infection, you collected DNA from cells that did not die. Can you distinguish between cells that did not become infected vs cells which became infected but did not die? Some microscopy to check presence / absence of the parasite in the surviving cells would have been interesting. Do you have any material remaining with which you can estimate the level of infection in the surviving cells? I find the subtitle “Genetic screen reveals genes required for infection and host response” a little misleading as I do not think you distinguish between these aspects. I don’t understand from your data whether knock out of your 35 top hits blocks invasion of the parasite, blocks progression of parasite life cycle development, or suppresses the host response (inflammatory? cell death?) to parasite infection. Please comment.

6. A follow on from point 5 – I suggest that you do a more in depth characterization of selected candidates (in particular those from the interferon pathway) and analyse in vitro at which point parasite invasion / development / egress is blocked. Monoclonal Cas9 knock outs would be great, but you can probably also obtain this info from your siRNA knock downs.

7. Fig S1B/C. The cell viability in the HCT-8 transduced with scrambled siRNA is decreased upon infection, but I would have expected a much more significant decrease. Were you expecting this result? Did you infect the cells with a 90% kill dose, like in the screen? Why was viability not assessed at 72 hours post infection (particularly considering the significant upregulation of ISG transcripts at 72h).

8. Figure 4: Here mouse models lacking IFN Type 1 receptor (ifnar -/-) or IFN type III receptor (Il28ra-/-) were used. Loss of type 1 receptor resulted in a reduction in oocyst shedding, while loss of type III interferon signaling resulted in an increase in parasite shedding. Data from the knock out mice was supported nicely with antibody blocking experiments. Together this showed that interferon type III signaling provides a protective response in early infection. It would help to tie this observation back to the data from the genome wide screen here, where you showed that losing interferon-signaling related genes blocked cell death in vitro. I must admit I got a bit lost here and would appreciate some more clarity in the discussion.

9. Figure 4: Parasite load was increased in mice lacking the type 1 interferon receptor (Ifnar-/- mice). This is a surprising observation. What is your explanation for this?

Reviewer #3: 1. Regarding the protective role of type 3 IFNs during cryptosporidiosis: The data presented present an apparent contradiction between the initial screen, showing several genes involved in the IFNs signaling pathway as important determinants of host cell death following infection, and the subsequent experiments showing the protective role of type III IFNs during cryptosporidiosis. If deleting these IFN signaling genes (i.e. STAT1/2, TYK2, IFNAR, IL10RB, IFNLR1, JAK1 and IRF9) strongly increased survival of host cells in vitro, how do the authors explain the protective role of this same pathway in vivo? In the discussion of this paper, they suggest that these observations might be the result of IFNs (type I and III in this instance) inducing cell death following their upregulation by infection. Unfortunately no support is provided for this hypothesis. It would be important to test whether administration of type 3 IFNs at the level seen in cells following infection are capable of inducing any cell death program – or if this outcome also requires infection.

2. Regarding the type 1 IFNs results: How do you explain the growth facilitating effect of type I IFNs observed in vivo (lower burden in Ifnar mice compared to WT in Fig 4C) – which could be consistent with the results of the in vitro screening? It seems that the authors focused their efforts on type III IFNs since the qPCR assessment of gene expression after infection showed almost no increase in expression of IFNB (Fig 2E). However, of the genes found significantly enriched in the screen, only the IFNLR subunits are not part of the type 1 IFNs pathway, and one type 1 IFNs receptor subunit was included. It would be beneficial to examine KO cells lines to decipher the role of each one of these cytokines during infection in vitro. Does loss of production of either type I or type III IFN, or their respective signaling receptors, protect cells against death due to infection?

3. Regarding the role of TLR3 in inducing the expression of type 3 IFNs following infection with Cp: In the initial testing of PRRs (Fig 5A), dsRNA did not induce the expression of IFNλ. In contrast, the authors propose that dsRNA – possibly originating from parasite viruses – is recognized by TLR3 to induce an immune response. However, it is hard to understand why these two different forms of dsRNA would have such different outcomes. Do you have any insight about how intraparasitic virus-derived dsRNA could end up exposed to endosomal TRL3?

Furthermore, you discuss the role of type 1 IFNs expression induced downstream of TLR3 activation in leishmaniasis, leading to disease exacerbation. In contrast, type I IFN does not seem to be strongly stimulated by C. parvum, despite evidence for signaling through TLR3. Might the differences observed be due to the fact that the two types of target-cells are widely different (IECs in cryptosporidiosis vs MΦs in leishmaniasis), or be due to some parasite-induced type 1 IFNs-inhibition mechanism? In this regard, you did measure IFNβ expression in HCT-8 cells following lipofection with poly(I:C) (Fig S2) which also activates TLR3.

Reviewer #4: Major comments:

1. The conclusion on line 263 that “…induction of the interferon pathway is exclusive to infected Neon positive cells” is not well supported by the data and the authors should consider revising. As I understand the experiment is comparing infected to bystander rather than to uninfected, so it could be that the bystanders are responding vs uninfected, but with reduced magnitude compared to infected. Additionally, it would be interesting to know whether any IFN or other gene signatures are present among the large number (466) of downregulated genes in infected vs uninfected.

2. Infection data in figures 4 and 5 need statistical analyses to back up conclusions regarding fold-change of AUC stated in the text. Conclusions drawn in text appear to be well-supported by data, but clarity in results of statistical tests are required.

3. Mouse phenotypes (especially gut phenotypes) can be impacted by microbiota or other environmental factors and controlled by use of littermates or cohousing. From methods, it seems that littermate mice or cohousing were not part of the experimental design (with perhaps exception of the Vil-cre/Stat1flox experiment?). Authors should clarify whether littermates were used or if co-housing of knockout with WT was employed for any of the experiments.

**Part III – Minor Issues: Editorial and Data Presentation Modifications**

Reviewer #1: Minor points:

(1) Author Summary, lines 52-53, the meaning of IFN-λ “induction likely preceding IFN-γ” is unclear, as a careful comparison of temporal dynamics was not carried out in the study.

(2) A standard CRISPR/Cas9 knockout screen was carried out in the study. Fig. 1A-D is not necessary and does not add to the paper.

(3) Top up-regulated and downregulated genes need to be highlighted in Fig. 2A and 3H. Otherwise, the graphs are not informative.

(4) Fig. 2E and F, IFN-β expression was not induced by Crytosporidium infection. What about the different IFN-α isoforms? Can the authors comment on why IFNAR2 was identified in the screen if HCT-8 cells do not produce type I IFNs?

(5) ISGs upregulated in RNA-seq data in Fig. 3H need to be validated by QPCR.

(6) Genetic background of mice and knockout mice used in Fig. 5 should be labeled in the graph.

(7) Fig. S3, immunohistology data of intestinal sections should be provided here.

(8) The complete analysis of CRISPR screening data needs to be provided in S3 Table.

(9) Proper statistical analyses are needed for all column graphs throughout the manuscript.

Regarding RNA-seq:

(1) What percent of reads aligned to the human genome? What percent to the cryptosporidium genome? The discrepancy in IFNL transcripts quantified with RNA-Seq and by qPCR may be explained by this.

(2) What type of transcript was considered a differentially expressed gene: protein-coding, noncoding, small RNAs, etc?

(3) The color-coding of the volcano plots in Figure 2A and Figure 3H are confusing: typically, a non-black color is given to all differentially expressed genes in a volcano plot while those that are not DEGs are black. It isn’t clear, unless reading the legend, that only genes enriching to a particular ontology term are colored red.

(4) The enrichments and genes for the 48-hour results are provided in the supplement with the text focusing on upregulated gene enrichment. It would be interesting to comment on what is downregulated. Are genes related to GAG and GPI biosynthesis included in any of these enrichments?

(5) Please be consistent when referring to transcriptomics: the manuscript alternates between “RNAseq” and RNA-Seq.”

Reviewer #2: Point out naming convention for the interferon type III receptor (Il28ra encodes for IFNLR1) to save confusion to non-interferon specialists.

Line 297: briefly comment on the phenotype of the ifng-/- mice (i.e. no interferon response at all?)

Line 305: in contrast to mock injected mice….. no mortality. Can you show this data? How many mice died and on which day of infection?

Line 361: Typo “Papillomavirus”

Line 431: Typo “TL3”

Reviewer #3: * Fig 4B: the D2 time-point shows a very high SE. Is this figure aggregating the results from the 2 replicates or just from 1 experiment? In this latter case, it would mean that only 1 of 2 mice displayed a detectable IFNλ level (I suppose that the dotted line represents the limit of detection), making the present representation a bit misleading in my opinion.

* Fig 4G: In the text, you state (discussion): “Consistent with this idea treatment of Rag2 -/- Il2rg -/- was repeatedly less effective than that of Rag2 -/- alone (Fig 4G), suggesting a potential role for IFNλ in promoting IFNγ production in NK/ILCs”. This is contradicted in the result section: “BL6, Rag2 -/- , and Rag2 -/- Il2rg -/- mice were infected and treated with 1μg of Ifnλ2. This resulted in comparable reduction of parasite shedding (BL6: 2.4-fold, Rag2 -/- : 4.4-fold, Rag2 -/- Il2rg -/- : 2-fold, Fig 4G), again suggesting that the benefit of IFNλ treatment does not require immune cells, but largely rests on an enterocyte intrinsic response”. Furthermore, the choice of representative figures for these 3 replicates is a bit misleading, presently showing BL6 treatment as highly effective, whereas, per the results presented in the text, it is not much more effective than Rag2 -/- Il2rg -/- treatment.

* Fig 4 legends: Many of these examples report showing a “representative example of several biological replicates”. The sample sizes are quite small here. It would be preferable to show aggregate data with combined statistical analysis for all biological replicates – or alternatively show the other biological replicates in the supplement. If you chose this option, in the text, please specify if the same statistical findings were seen in all replicates or only some.

* Fig 5E: the results section states: “We found the production of IL-18, an enterocyte derived cytokine induced by Cryptosporidium infection, [46, 47] to be intact in the absence of TLR3 (Fig 5E)” but, in fact, these results show the IL-18 expression to be increased in TLR3 KO mice by more than 2 fold in a consistent way. Is it possible that the increased parasite burden due to TLR3 ablation is responsible for this increased expression of IL-18? Did you check if this increase is correlated with an increase in IFNγ production in these mice? If so, this result might contradict the claim that there is “a potential role for IFNλ in promoting IFNγ production in NK/ILCs”?

Reviewer #4: Minor comments:

4. Fig 3H. In this representation the ISGs (red) are identified by Interferome DB rather than the curated REACTOME gene set used in earlier figure 1A-B. It would be nice to keep the signature used for ISGs consistent between these gene expression studies in different figures.

5. Lines 278-280. The type I interferon receptor is a heterodimer of IFNAR1 and IFNAR2, but text implies that there is only one IFNAR gene. Authors should clarify this point in the text, and be clearer about which knockout was used (Ifnar1 as per methods).

6. Authors may want to use the updated ncbi gene name Ifnlr1 (rather than Il28ra) when referring to the knockout mouse because this is more clearly matching the IFNL cytokine nomenclature.

7. Authors should indicate the specific strain of Ifnlr1 knockout in the methods. If it was obtained from BMS, I presume it the Ifnlr1tm1Palu line originally generated by Ank et al. 2008 (10.4049/jimmunol.180.4.2474).

8. Lines 1109-1110. This panel description is confusing to read. Maybe add parentheses around mouse knockout names?

9. Labels to identify bars in panels 4G and 4H are absent. I presume that open bars are IFNL treated as in panel 4F, but this should be clarified in the legend.

10. Line 297. The authors should include a rationale for switching to use ifng-/- mice for the IFNL therapeutic experiments. Was IFNL treatment ineffective in WT?

11. Line 304-305. The authors indicate a difference in mortality, but should also report numbers if they wish to draw a conclusion about about mortality difference between groups.

PLOS authors have the option to publish the peer review history of their article (what does this mean?). If published, this will include your full peer review and any attached files.

Reviewer #1: No

Reviewer #2: No

Reviewer #3: No

Reviewer #4: No
---

## [Editor Report · Decision Letter 1]

11 Apr 2022

Dear Dr Striepen,

We are pleased to inform you that your manuscript 'A genetic screen identifies a protective type III interferon response to Cryptosporidium that requires TLR3 dependent recognition' has been provisionally accepted for publication in PLOS Pathogens.

Best regards,

Philipp Olias

Associate Editor

PLOS Pathogens

Vern Carruthers

Section Editor

PLOS Pathogens

Kasturi Haldar

Editor-in-Chief

PLOS Pathogens

orcid.org/0000-0001-5065-158X

Michael Malim

Editor-in-Chief

PLOS Pathogens

orcid.org/0000-0002-7699-2064
---

## [Editor Report · Acceptance letter]

13 May 2022

Dear Dr Gibson,

We are delighted to inform you that your manuscript, "A genetic screen identifies a protective type III interferon response to *Cryptosporidium* that requires TLR3 dependent recognition," has been formally accepted for publication in PLOS Pathogens.

Best regards,

Kasturi Haldar

Editor-in-Chief

PLOS Pathogens

orcid.org/0000-0001-5065-158X

Michael Malim

Editor-in-Chief

PLOS Pathogens

orcid.org/0000-0002-7699-2064